# Structural basis for late maturation steps of mitochondrial respiratory chain complex IV within the human respirasome

Minh Duc Nguyen [1,2,6], Ana Sierra-Magro [3,6], Vivek Singh [1], Anas Khawaja [1], Alba Timón-Gómez [3], Antoni Barrientos [3,4,5] ✉ & Joanna Rorbach [1] ✉

The mitochondrial respiratory chain comprises four multimeric complexes (CI-CIV) that drive oxidative phosphorylation by transferring electrons to oxygen and generating the proton gradient required for ATP synthesis. These complexes can associate into supercomplexes (SCs), such as the $CI + CIII_2 + CIV$ respirasome, but how SCs form, by joining preassembled complexes or by engaging partially assembled intermediates, remains unresolved. Here, we use cryo-electron microscopy to determine high-resolution structures of native human $CI + CIII_2 + CIV$ late-assembly intermediates. Together with biochemical analyses, these structures show that respirasome biogenesis concludes with the final maturation of CIV while it is associated with fully assembled CI and $CIII_2$. We identify HIGD2A as a placeholder factor within isolated and super-complexed CIV that is replaced by subunit NDUFA4 during the last step of CIV and respirasome assembly. This mechanism suggests that placeholders such as HIGD2A act as molecular timers, preventing premature incorporation of NDUFA4 or its isoforms and ensuring the orderly progression of pre-SC particles into functional respirasomes. Since defects in CIV assembly, including NDUFA4 deficiencies, cause severe encephalomyopathies and neurodegenerative disorders, understanding the molecular architecture and assembly pathways of isolated and supercomplexed CIV offers insight into the pathogenic mechanisms underlying these conditions.

The mitochondrial respiratory chain (MRC) consists of four multimeric enzymatic complexes (CI to CIV) and two mobile electron carriers, ubiquinone (Q) and cytochrome *c* (cyt *c*), which act in concert to drive the transfer of electrons from substrates such as NADH and succinate to molecular oxygen[1,2]. Electron transfer along the MRC is coupled with the extrusion of protons through CI (NADH:ubiquinone oxidoreductase), CIII (ubiquinol-cytochrome *c* oxidoreductase), and CIV (cytochrome *c* oxidase) across the mitochondrial inner membrane. This process establishes a proton gradient that $F_1F_o$-ATP synthetase

(Complex V) uses to synthesize ATP through oxidative phosphorylation (OXPHOS).

Given the central role OXPHOS plays in cellular energy metabolism, deficiencies in the enzymes catalyzing these processes contribute to human diseases. This impact extends to both rare inherited mitochondrial disorders, including Leigh syndrome and cardioencephalomyopathies, and common age-associated diseases[3]. Therefore, a comprehensive understanding of the assembly, regulation, and functioning of these complexes is crucial for unraveling the mechanisms

[1]Department of Medical Biochemistry and Biophysics, Karolinska Institutet, Stockholm, Sweden. [2]Faculty of Pharmacy, Phenikaa University, Hanoi, Vietnam. [3]Department of Neurology. University of Miami Miller School of Medicine. 1600 NW 10th Ave. Miami, Miami, USA. [4]Department of Biochemistry and Molecular Biology. University of Miami Miller School of Medicine. 1600 NW 10th Ave. Miami, Miami, USA. [5]The Miami Veterans Affairs (VA) Medical System. 1201 NW 16th St, Miami, USA. [6]These authors contributed equally: Minh Duc Nguyen, Ana Sierra-Magro. ✉e-mail: abarrientos@med.miami.edu; joanna.rorbach@ki.se

underlying cellular energy metabolism and its implications in various health conditions[4]. Despite the advances resulting from the extensive work performed in many labs worldwide, fundamental questions remain open regarding the assembly and organization of the MRC complexes and their impact on function and metabolic adaptation.

The MRC is organized as individual complexes that coexist with higher-order assemblies known as supercomplexes (SCs)[5,6]. In mammalian mitochondria, the SCs are formed by the proton-pumping enzymes −CI, CIII, and CIV− in various stoichiometries[7–9]. They include CI + CIII$_2$, CIII$_2$ + CIV, and SCs CI$_n$ + CIII$_n$ + CIV$_n$, termed respirasomes because they are capable of NADH:O$_2$ oxidoreduction in vitro. Recent in situ cryo-electron microscopy has revealed four major respirasome organizations: CI + CIII$_2$ + CIV, CI + CIII$_2$ + CIV$_2$, CI$_2$ + CIII$_2$ + CIV$_2$, and CI$_2$ + CIII$_4$ + CIV$_2$[10]. Although the physiological roles of respirasomes remain controversial, they have been extensively characterized both structurally and biochemically[10–14], particularly the simplest and most predominant form, CI + CIII$_2$ + CIV, shedding light on electron transfer, proton pumping, and regulatory mechanisms[2,15].

The functional relevance of SCs may be intricately tied to the regulatory mechanisms governing their biogenesis, which are not yet fully elucidated. Two contrasting models have been proposed to explain SCs' biogenesis and their functional coexistence with individual complexes. The plasticity model posits that SCs form only after the individual complexes are fully assembled[7], while the cooperative assembly model proposes that SC assembly begins with intermediates of individual complexes[16,17]. Emerging evidence supports the second model, showing that partially-assembled complexes can interact in the context of SCs[15,18–20]. Additionally, it has been hypothesized that SCs may serve as a platform for coordinated complex biogenesis regulation, influencing the assembly and stability of the individual complexes[18,21,22].

Recent investigations have shown that CIV subassemblies interact with CI and CIII in SCs and have suggested that alternative assembly pathways coexist to form "free" versus "respirasome-bound" CIV[19,23]. CIV assembly is believed to be a modular process where the catalytic mt-COX1, mt-COX2, and mt-COX3 subunits, encoded in the mitochondrial genome, form preassembly modules with several of the eleven nucleus-encoded subunits forming the complex[24,25]. The maturation of each catalytic subunit and the formation of modular assemblies are assisted by more than thirty assembly factors[25]. The search for accessory factors assisting the assembly of CIV-containing respirasomes identified the Cytochrome $c$ Oxidase subunit isoform 7A2-like COX7A2L[26–29]. While COX7A2L was initially considered essential for CIV binding in both CIII$_2$CIV and respirasomes[27,29], further structural evidence suggests it is needed for CIII$_2$CIV formation rather than the respirasome itself[30]. However, complexome profiling and structural studies have identified two distinct CI + CIII$_2$ + CIV respirasome classes containing either COX7A2L or COX7A2 isoforms, preferentially assembling depending on metabolic cues[31,32]. On the other hand, the Hypoxia Inducible Gene Domain family protein HIGD2A binds to and promotes the association of the mt-COX3 module into both individual CIV and respirasomes[23,33]. Additionally, it has been proposed that HIGD2A promotes the formation of respirasome assembly intermediates by coordinating the association of CIV assembly modules, revealing divergent assembly pathways for individual and supercomplexed CIV[19,34]. The heterogeneity of SC populations coexisting in the inner mitochondrial membrane (IMM) and the challenge of separating SC assembly intermediates from their mature forms for structural studies has complicated assigning the roles of different factors in the SC assembly processes.

In this study, we isolated the human respirasome CI + CIII$_2$ + CIV from human embryonic kidney (HEK293) cells and solved high-resolution structures of its constituent complexes using cryo-electron microscopy (cryo-EM). We identified the late-stage respirasome assembly intermediates that capture the final steps of CIV assembly after CI and CIII are fully matured. Detailed structural and biochemical analyses revealed that HIGD2A binds to CIV, serving as a placeholder for the final incorporation of the NDUFA4 subunit, which completes CIV and finalizes respirasome biogenesis. Our findings offer critical insights into the temporal dynamics and molecular intricacies of CIV assembly, both as an isolated complex and within the context of the respirasome.

## Results

### Determination of human respirasome structure by cryo-EM

To gain insight into the assembly of CIV, we engineered a C-terminal FLAG tag on the CIV biogenesis factor COX14, which has previously been shown to associate with COX1 during its translation and CIV maturation within the IMM[35,36]. Mitochondrial membranes were solubilized with lauryl maltose neopentyl glycol (LMNG), followed by affinity purification and cryo-EM analysis of the eluates (see the "Methods" section). We detected abundant 2D classes representing unique views of the respirasome SC (Supplementary Fig 1), which were pooled for 3D reconstruction. This yielded a map of CI + CIII$_2$ + CIV respirasome with an overall resolution of 3.1 Å (Fig. 1A). The membrane domains of the complexes were visualized with a bound detergent belt displaying a slight curvature, consistent with recently reported SC structures[10,37]. Despite extensive data processing and map analysis, we were unable to identify any additional density corresponding to COX14, the bait protein, most likely due to its transient interaction and subsequent loss during post-immunoprecipitation processing.

In the initial map after homogeneous refinement, the density for CIV appeared weaker than CI and CIII, indicating greater conformational flexibility, in line with previous studies[11–14,38]. This observation was further supported by 3D variability analysis using the mask covering both CIII and CIV (Supplementary Movie 1). Local refinement on CI, CIII$_2$, and CIV significantly improved their respective resolutions to 2.9 Å, 2.77 Å, and 3.1 Å (Supplementary Fig. 1). In the following sections, we describe the structures of each complex, with a particular focus on the resolution of several CIV assembly intermediates within these SCs.

### Structure of human complex I at 2.7 Å resolution

A final focus map on CI built from a total of 236,346 particles yielded 2.66 Å resolution with consistent local resolution (Supplementary Figs. 1, 2, Supplementary Table 1). We identified distinct electron densities for flavin mononucleotide (FMN) and dihydro-nicotinamide-adenine-dinucleotide phosphate (NADPH) molecules, as well as Fe−S clusters, but not for any NADH molecule (Supplementary Fig. 3A). Based on both the overall enzyme conformation and structural features of key components of CI (Supplementary Fig. 3A, B), we concluded that CI is in an active resting state, consistent with previous findings[39–43]. Therefore, we will not discuss the open or inactive states of the complex, as these have been thoroughly examined in earlier studies[42,43] and reviewed recently[44,45]. Additionally, we observed ambiguous broken densities throughout the ubiquinone-binding channel (Supplementary Fig. 3A), reminiscent of those observed in previous active/closed structures of mammalian CI that also lacked a defined bound ligand[39,40,43].

The improved resolution compared to earlier models of human CI[14] allowed us to build a more accurate model (Supplementary Table 2). We modeled fourteen common phospholipids, including 6 cardiolipin molecules and 8 molecules of 1,2-dioleyl-sn-glycerol-3-phosphothanolamine (PEE), primarily located at the interface between CI and CIII$_2$ (Supplementary Fig. 4A). Additionally, the nucleotide in subunit NDUFA10 was modeled as a Mg$^{2+}$-bound 2′-deoxyguanosine-5′-triphosphate (Supplementary Fig. 3C, **right**), in agreement with recent biochemical and structural data from other species[46,47]. Notably, while previous studies reported the demethylation of mouse CI NDUFS2-Arg85[42,47,48], our analysis revealed the corresponding human CI residue, Arg118, as monomethylated (Supplementary Fig. 3C, **left**). Interestingly, we found the phosphorylation at the Ser58 residue of the

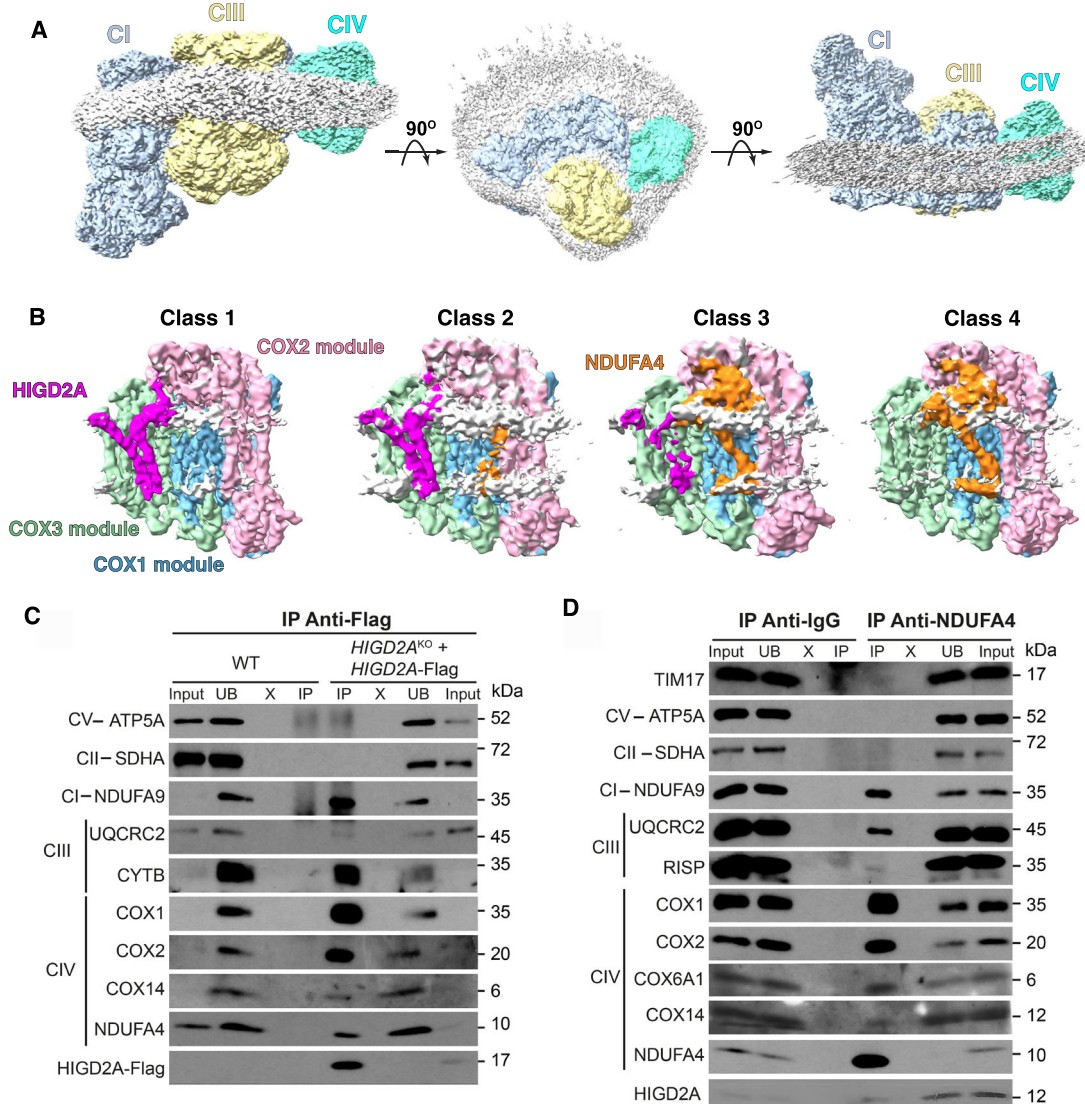

**Fig. 1 | Overall map of the respirasome complex and distinct classes of complex IV within the respirasome. A** Different views of representative maps of the respirasome obtained in this study. The maps of complex I (CI), complex III (CIII), and complex IV (CIV) are shown as light blue, yellow, and cyan, respectively. The detergent belt is shown as light gray. The contour levels of the maps (σ) are 0.15. **B** Different classes of CIV obtained after 3D classification, focusing on the COX3 module. The densities for proteins from the assembly COX1 module (COX1, COX4i1, COX7B, COX7C and COX8A), COX2 module (COX2, COX5A, COX6B1, and COX6C) and COX3 module (COX3, COX5B, COX6A1 and COX7A2) are shown in blue, pink, and green, respectively. The densities representing HIGD2A and NDUFA4 are shown in magenta and orange, respectively. The detergent micelles

are shown as light gray. The orientation of CIV relative to CICIII₂ is the same as the left panel in (**A**). The contour levels of the maps (σ) are 0.3. **C** Representative western blot images of HIGD2A-interacting proteins. Immunoprecipitation with anti-FLAG agarose beads was performed for mitochondrial lysates from *HIGD2A*-KO cells expressing HIGD2A-FLAG and WT cells as a control. Lanes: input; unbound (UB); washes (X); elution (IP). This experiment was repeated four times with similar results. **D** Immunoprecipitation of NDUFA4-interacting proteins with protein A agarose beads conjugated to anti-NDUFA4 antibody. Immunoprecipitation using anti-IgG was used as a control. This experiment was repeated four times with similar results. Source data are provided as a Source Data file.

NDUFS7 subunit, which has not been previously reported (Supplementary Fig. 3C, **middle**). Taken together, this finding indicates that post-translational modification patterns may vary depending on species or tissue type.

### Structure of human complex III at 2.5 Å resolution

To date, only one structure of human CIII has been resolved, with an overall resolution of 3.4 Å[14]. In the current study, we obtained a final focus map on CIII from a total of 213,731 particles, resulting in the CIII structure at a resolution of 2.5 Å for most regions, except for the dynamic Rieske iron-sulfur protein (RISP) (Supplementary Figs. 1, 2, Supplementary Table 1). This high-resolution map enabled us to build accurate models for most of the proteins within the complex

(Supplementary Fig. 4B, C, Supplementary Table 3). CIII exists as a homodimer, in which we unambiguously modeled two metal centers per monomer, including two heme molecules (heme $b_H$ and $b_L$) in MT-CYB and one heme $c_1$ molecule in cyt $c_1$ (CYC1) (Supplementary Fig. 4B, **insertion**). Additional weak densities were observed at the two ubiquinone (UQ) binding sites, which we modeled as Q10 molecules, although we were unable to determine their redox states (Supplementary Fig. 4B, **insertion**). Notably, our map did not show any additional densities at the cyt c binding site, which were previously reported in human respiratory megacomplex CI₂CIII₄CIV₂[14]. Additionally, we did not detect a clear density corresponding to the Fe-S cluster (Supplementary Fig. 4B, **insertion**), likely due to the conformational heterogeneity of RISP within CIII, as previously reported[10,13,14].

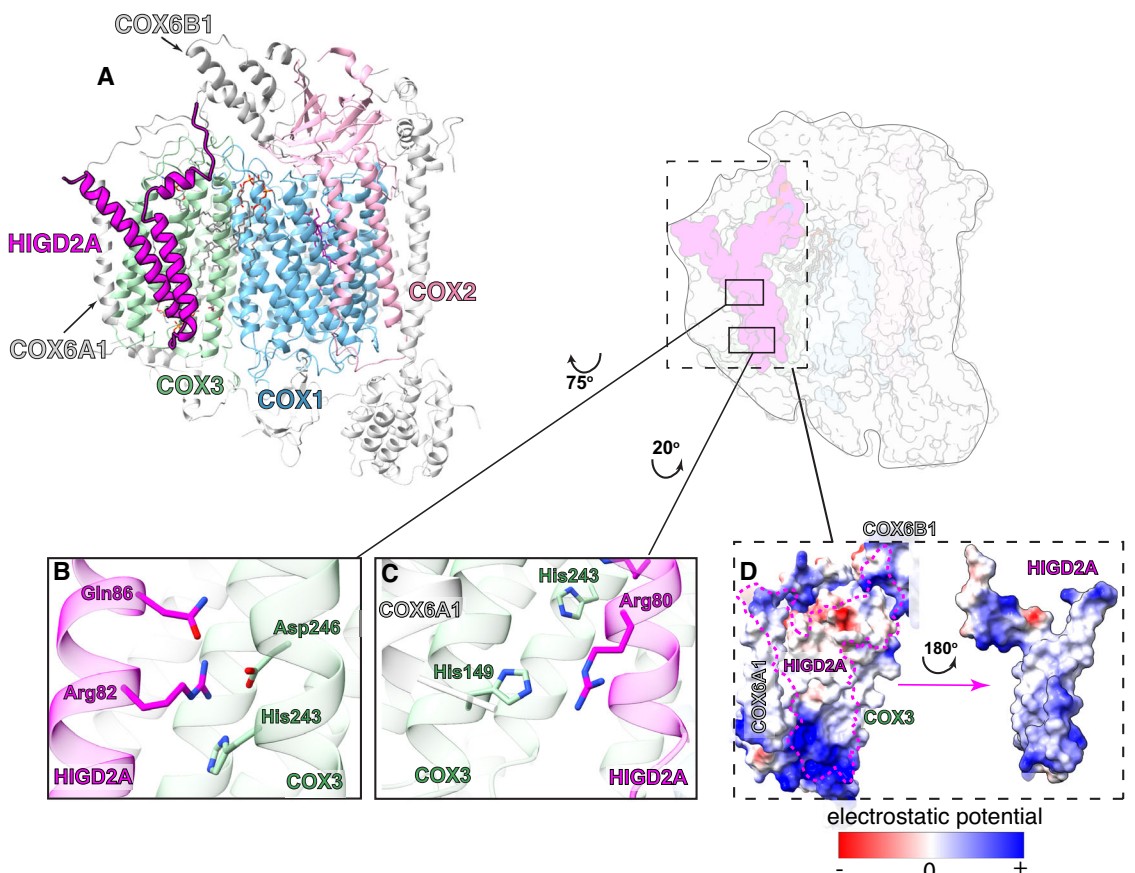

**Fig. 2 | Structural features of HIGD2A-bound complex IV. A** Overview of the complex. The core subunits or mtDNA-encoded proteins COX1, COX2, and COX3 are shown as light blue, pink, and light green, respectively. The HIGD2A is shown as magenta, the lipid molecules are in gray, and the heme molecules are in purple. The rest of the nucleus-encoded subunits are shown as light gray. The COX6A1 protein is in close contact with HIGD2A, as indicated. **B**, **C** The close-up views of interactions between residues (Arg80, Arg82, and Gln86 from the QRRQ motif of HIGD2A with residues (His149, Asp246, and His250) of COX3. **D** The electrostatic potential surface of the binding pocket for HIGD2A (dashed magenta), formed by COX3, COX6A1, and COX6B1, is shown on the left panel. The electrostatic potential surface of HIGD2A, which makes contact with COX3, COX6A1, and COX6B1, has been shown in the right panel. Electrostatic surface potentials are colored red and blue for negative and positive charges, respectively, and white color represents neutral residues.

## HIGD2A- and NDUF4A-bound intermediate structures of complex IV

Local refinement of CIV revealed high flexibility, suggesting structural heterogeneity. A 3D variability analysis, using a mask generated during local refinement, revealed the presence of additional proteins within a cleft formed by COX1, COX2, COX3, and COX6A1. Interestingly, previous crystal structures of the homodimeric CIV[49] suggest that this cleft plays a critical role in mediating interactions between two CIV units.

To investigate this further, we generated a mask covering the COX3 module and extending toward the cleft. As a result, the subsequent 3D classification revealed four distinct classes with varying density patterns surrounding the cleft (Fig. 1B).

Specifically, class 1 and class 4 exhibited distinct extra densities corresponding to HIGD2A and NDUFA4, respectively. Class 2 displayed density for HIGD2A, along with additional density near helix 2 of COX2, indicating the impending binding of NDUFA4 to CIV. Class 3 revealed clear extra density for NDUFA4 and a weaker density at the HIGD2A binding region, suggesting the displacement of HIGD2A upon NDUFA4 recruitment. Together, these structural analyses suggest a dynamic interplay between HIGD2A and NDUFA4 during the assembly of CIV, with HIGD2A likely being released from CIV concurrently with the incorporation of NDUFA4 (Fig. 1B).

To biochemically confirm the simultaneous presence of HIGD2A and NDUFA4 in a subset of particles, we performed FLAG-immunoprecipitation assays using wild-type (WT) HEK293T cells and HIGD2A-KO cells expressing HIGD2A-FLAG. Additionally, we performed immunoprecipitation for NDUFA4 in WT cells using anti-NDUFA4-conjugated beads. In both cases, we detected components of CI, CIII, and CIV (Fig. 1C, D), confirming that both HIGD2A and NDUFA4 interact with CIV within the context of the respirasome. Additionally, small substoichiometric amounts of NDUFA4 were co-immunoprecipitated with HIGD2A-FLAG (Fig. 1C), and similarly, small amounts of HIGD2A were co-precipitated with NDUFA4 (Fig. 1D). This observation supports the cryo-EM data, indicating that only a minor fraction of these two proteins co-exists within the same CIV assembly intermediates.

## Structural features of HIGD2A-bound complex IV

Local refinement of class 1, focusing on complex IV, yielded a map at 2.93 Å resolution (Supplementary Fig. 1), exhibiting higher resolution at the core subunits and lower resolution at the periphery. An additional density corresponding to a polypeptide with 2 transmembrane helices was observed near COX3 and COX6A, but at a relatively lower local resolution in the range of 3.5 to 6 Å (Supplementary Fig. 2, class 1 of complex IV). We modeled HIGD2A into the extra density in class 1, achieving sufficient resolution to assign most amino acid side chains from Asn31 to Pro106 within the distal C-terminal domain (Fig. 2A, Supplementary Fig. 5A).

HIGD2A and its yeast functional homolog, Rcf1, belong to the HIG1 family type 2 isoforms due to their conserved $(Q/I)X_3(R/H)XRX_3Q$ (QRRQ) motif (Supplementary Fig. 6A, B), which is necessary for

binding COX3 and regulating modular CIV assembly[23,50]. The structural model revealed specific interactions of the residues Arg80, Arg82, and Glu86 of the conserved QRRQ motif, with highly conserved residues His149, Asp246, and His250 of COX3 (Fig. 2B, C). Within the IMM, HIGD2A engages in hydrophobic and Van der Waals interactions with COX3 and COX6A1 (Fig. 2D). For comparison, a previous structural study in yeast (S.cerevisiae) identified Rcf2 (PDB: 6T0B), another type 2 isoform of the HIG1 family (Supplementary Fig. 6C, **middle panel**), bound to the mature $CIII_2 + CIV_{1-2}$ SCs, occupying a pocket formed by CIV subunits Cox1, Cox3, Cox12, and Cox13 (ref[51]). Recently, the structure of Rcf2-bound CIV (PDB: 8C8Q[52]) from yeast (S.pombe) revealed specific interactions between the fully conserved Arg143 and Gln147 of the QRRQ motif and Asp254 of COX3 (Supplementary Fig. 6C, **right panel**). This interaction with mature CIV closely resembles that of HIGD2A, even though Rcf2 is described as modulating CIV function rather than its assembly[53], and it is considered the functional homolog of human HIGD1A, a type-1 HIG1 family member (Supplementary Fig. 6A, B). Although HIGD2A's functional homolog is Rcf1, it has been suggested that Rcf1 binds CIV in a similar manner to Rcf2[52], which may explain the parallels between HIGD2A binding to CIV described here and the previously characterized interaction of Rcf2 with CIV, despite their distinct functions. Nonetheless, some differences are observed between HIGD2A and Rcf2 binding to CIV. For instance, the yeast SC structure revealed the C-terminal α-helix of yeast Rcf2 extending into the IMS, to form electrostatic interactions with Cox12[51]. In contrast, HIGD2A lacks this C-terminal extension (Supplementary Fig. 6A), and we did not detect any additional interactions between the IMS-protruding regions of HIGD2A and COX6B1, the human homolog of yeast Cox12. Likewise, interactions observed between Rcf2 and Cox13[51], which occupies a position analogous to COX6A1 in human CIV, were absent in our structural model.

In our HIGD2A-bound structure, we identified a well-defined cardiolipin (CDL) molecule with its head group inserted into a deep pocket formed by COX1, COX3, and the IMS side of COX6B1 (Supplementary Fig. 5C). This CDL likely plays a role in maintaining the structural integrity and function of the complex[54]. Additionally, three phosphatidylethanolamine (PEE) molecules were previously reported within COX3 in both cryo-EM and crystal structures of human CIV[54]. In our structure, we observed an additional lipid molecule, assigned as PEE, bound to COX3, extending toward the surface and interacting with COX7A2 (Supplementary Fig. 5E). This binding site was previously shown to be occupied by a CDL molecule in crystal structures of both dimeric[55] and monomeric bovine CIV[56].

### Structure of NDUFA4-bound complex IV

Local refinement of class 4 led to a 3.2 Å resolution reconstruction. Overall, the map is in agreement with a previous cryo-EM structure of human CIV, resolved at 3.6 Å resolution, which identified NDUFA4 as the fourteenth subunit of CIV[54] (Fig. 3A, Supplementary Fig. 1 and Supplementary Fig. 5B). Although the local resolution of the map covering the NDUFA4 region is lower than 3.2 Å, ranging from 4 to 6 Å (Supplementary Fig 2, **class 4 of complex IV**), clear side chain densities are visible for several bulky residues, including Phe19, Tyr32 and Trp45 (Supplementary Fig. 5B). As expected, the N- and C-termini of NDUFA4 are located on the matrix side and IMS, respectively (Fig. 3A). The N-terminal region forms a short α-helix within the IMM toward the matrix, while the C-terminus is in close contact with COX6B1 in the IMS. The central region of NDUFA4 forms a transmembrane helix (TMH) that parallels the main TMHs of COX1 and is adjacent to TMH2 of COX2 (Fig. 3A). Interestingly, the presence of NDUFA4 creates a deep pocket formed by the main TMHs of COX1 and COX3, along with the C-terminal region of COX3 and the TMH of both NDUFA4 and COX6B1 within CIV. This pocket stabilizes the binding of the CDL molecule, described above for the HIGD2A-bound complex. As a

result, the density for the head of the CDL molecule is more clearly resolved in the presence of NDUFA4 compared to HIGD2A (Supplementary Fig. 5C, D).

To evaluate the functional consequences of NDUFA4 incorporation into CIV, we assessed its enzymatic activity in NDUFA4-KO cells. Consistent with previous reports[57,58], the absence of NDUFA4 led to an approximately 50% reduction in CIV activity, as determined by both spectrophotometric assays (Fig. 3B) and in-gel activity staining (Fig. 4H). This reduction correlated with a comparable (~50%) decrease in endogenous respiration in NDUFA4-KO cells relative to WT HEK293T cells (Fig. 3C, D).

In agreement with our structural studies, BN-PAGE analyses revealed that NDUFA4 comigrates with other CIV subunits in both isolated CIV and CIV-containing SCs in HEK293T and 143B human cell lines (Supplementary Fig. 7A). This comigration was also observed in an NDUFB7-KO CI mutant cell line, where CIV distribution is only secondarily affected (Supplementary Fig. 7A). NDUFA4 has been proposed as the final subunit added to CIV during assembly[59]. To investigate this, we analyzed the distribution of NDUFA4 in assembly intermediates using BN-PAGE in 143B cybrids with homoplasmic mutations in the mtDNA-encoded CIV core subunits COX1[60], COX2[61], or a nearly homoplasmic mutation in COX3[62] (Supplementary Fig. 7B), as these subunits form assembly modules[19,63]. NDUFA4 was not detected in any of the CIV subassemblies (Supplementary Fig. 7C) but was only associated with the small amount of fully assembled CIV present in the COX3-mutant cybrid cell line (Supplementary Fig. 7C). This suggests that NDUFA4 is indeed incorporated during the final steps of CIV assembly.

### Hierarchical incorporation of HIGD2A and NDUFA4 to CIV

Our structural data indicate that NDUFA4 and HIGD2A have partially overlapping binding sites on CIV, suggesting potential competition between these proteins for binding (Supplementary Fig. 8A). To elucidate their incorporation order, we analyzed the distribution and levels of HIGD2A- and NDUFA4-containing CIV in HIGD2A-KO and NDUFA4-KO cell lines (Fig. 4A, B, Supplementary Fig. 8B, C). In HIGD2A-KO cells, NDUFA4 levels were reduced to a similar extent as other CIV subunits. Conversely, in the absence of NDUFA4, HIGD2A redistributed from the previously described 50-kDa regulatory complex[23] to fully assembled isolated or supercomplexed CIV (Fig. 4B). Quantitative analysis of HIGD2A binding to CIV in NDUFA4-KO cells was performed by excising bands corresponding to holo-CIV and SCs from a first-dimension BN-PAGE gel, followed by second-dimension SDS-PAGE and immunoblotting (Supplementary Fig. 8B). The quantification revealed that, in the absence of NDUFA4, HIGD2A accumulated in CIV and SCs more than 3-fold and 2-fold, respectively (Supplementary Fig. 8C). These findings suggest that HIGD2A is stabilized on CIV when its release is not triggered by NDUFA4.

To further investigate the potential competition between HIGD2A and NDUFA4 for CIV binding, we overexpressed each protein in NDUFA4-KO, HIGD2A-KO, and wild-type (WT) cells. In WT cells, the proteins exhibited mutual exclusivity: overexpression of HIGD2A reduced NDUFA4 binding to CIV, and vice versa (Fig. 4C, D). In HIGD2A-KO cells, NDUFA4 overexpression partially restored CIV subunit levels, assembly, and activity (Fig. 4E, F). However, overexpression of HIGD2A in both WT and NDUFA4-KO cells had detrimental effects (Fig. 4G, H). These observations support a model for the final steps of CIV assembly within the respirasome, where HIGD2A facilitates the incorporation of the COX3 assembly module[23,33] and initially remains bound to CIV, acting as a placeholder. Subsequently, NDUFA4 displaces HIGD2A, completing the assembly of CIV and the respirasome assembly (Fig. 5).

### Discussion

Our cryo-EM reconstructions of the late stages of $CI + CIII_2 + CIV$ respirasome assembly and biochemical data refine the current

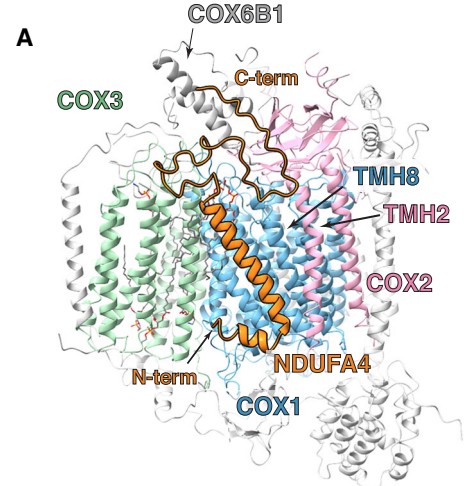

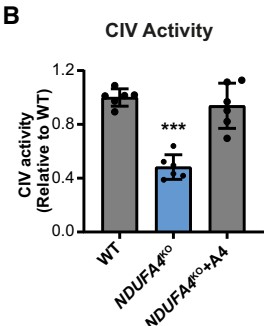
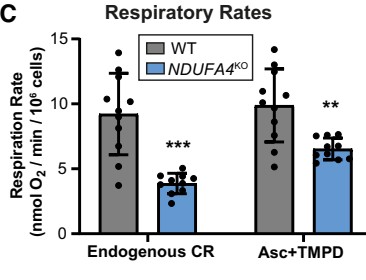
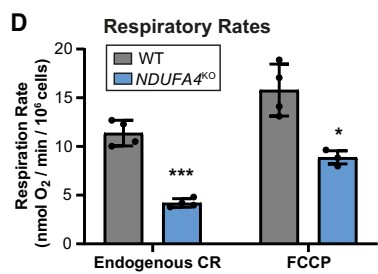

**Fig. 3 | Structural features of NDUFA4-bound complex IV. A** The structure of NDUFA4-bound CIV. NDUFA4 is colored orange, while the mtDNA-encoded enzymatic core subunits COX1, COX2, and COX3 are colored light blue, light pink, and light green, respectively. **B** Spectrophotometric measurement of complex IV activity in whole-cell extracts from HEK293T wild-type (WT), *NDUFA4*-KO, and KO cells reconstituted with *NDUFA4* (+A4). Enzymatic activity was expressed as nmol of cytochrome c oxidized per minute, normalized to protein content, and shown relative to WT values. The bar graph shows the mean ± SD from 6 replicates. One-way ANOVA followed by Tukey's post hoc test (two-sided). \*\*\**p* < 0.001,

*p* = 0.000004. **C, D** Polarographic analysis of KCN-sensitive endogenous cell respiration (CR) in the presence or absence of the cytochrome *c* electron donors ascorbate and TMPD (C) or the uncoupler FCCP (D). Respiratory activity was calculated as nmolO$_2$/min/10$^6$ cells. The bar graphs show mean ± SD from *n* = 10 (C) and *n* = 4 (D) independent experiments. Dots represent individual data points. Welch's *t*-test (two-sided). \**p* < 0.05, \*\**p* < 0.01, \*\*\**p* < 0.001. Left graph (C) *p* = 0.0002, right graph (C) *p* = 0.0027, left graph (D) *p* = 0.0007, right graph (D) *p* = 0.0112. Exact test statistics are provided in the Source Data Table.

understanding of the cooperative assembly pathways[17,19] of the respirasome complex (Fig. 5) and highlight an unexpected role HIGD2A plays in free and supercomplexed CIV biogenesis.

Previous studies had shown that HIGD2A interacts with the newly synthesized COX3 subunit to chaperone its modular assembly and incorporation into the isolated holoenzyme assembly line[23,33]. HIGD2A was not present in the fully assembled enzyme, indicating release upon assembly completion. In the absence of HIGD2A, the COX3-assembly module cannot form or becomes unstable, which results in decreased steady-state levels of its component subunits (COX3, COX6A1, COX6B1, COX7A2) and also NDUFA4[33], suggesting the late-assembly nature of this subunit, and the accumulation of the COX1- and COX1 + COX2-assembly modules[23].

Additionally, HIGD2A had been shown to form a 50-kDa complex with three nucleus-encoded subunits −COX4-1, COX5B, and COX6A1− which are components of the COX1, COX2, and COX3 assembly modules, respectively. This complex is proposed to coordinate the assembly of the three modules by releasing HIGD2A as needed for the assembly and incorporation of the COX3 module[23,64].

Notably, studies in 143B *COX2* cybrid cell lines revealed that a fraction of the 50-kDa HIGD2A complex (lacking COX6A1) can incorporate directly into the SC CI + CIII$_2$ and immediately recruit COX1 along with the COX2 and COX3 modules components, COX7C and

COX7A2, respectively[19]. This intermediate, termed "SC I + III$_2$ plus", is thought to facilitate the subsequent incorporation of COX2 and COX3 with their remaining module subunits, completing the assembly of supercomplexed CIV[19,23]. Importantly, HIGD2A has not been detected in the fully assembled CI + CIII$_2$ + CIV respirasome, suggesting its release upon assembly completion.

Our work illuminates the late stages of CIV assembly and their potential functional significance. We show that upon assembly and joining of the COX1, COX2, and COX3 modules, HIGD2A remains bound to the assembly intermediate, acting as a placeholder for NDUFA4. The incorporation of NDUFA4 is synchronized with the release of HIGD2A from CIV, completing the assembly of both isolated and supercomplexed CIV. It is tempting to speculate that NDUFA4 binding may influence proton transfer. In this scenario, oxygen reduction at the CIV catalytic center would proceed efficiently only in the fully mature holoenzyme, linking NDUFA4 binding to full functional activation of CIV, both as an individual complex and within the respirasome. Further investigation will be required to define the mechanistic link between NDUFA4 or HIGD2A incorporation and efficient oxygen reduction in isolated CIV and the respirasome.

NDUFA4 mutations have been identified in patients suffering from Leigh syndrome associated with CIV deficiency[57,58,65]. Our structural and biochemical data provide a mechanistic basis whereby

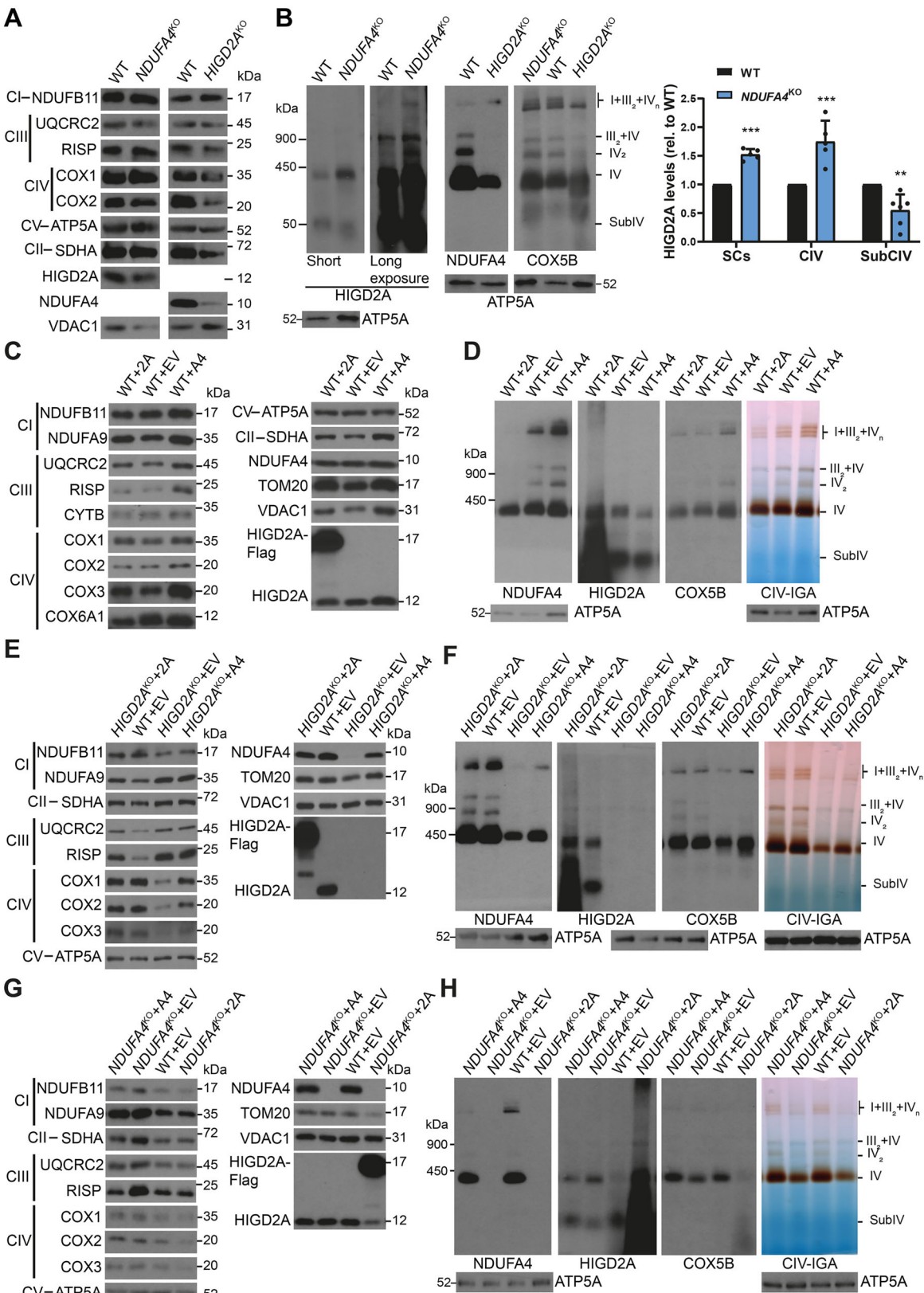

NDUFA4 loss disrupts the final maturation of CIV, impairing its activity and mitochondrial respiration without necessarily abolishing respirasome assembly. By delineating the temporal sequence and regulation of CIV assembly, our findings establish a direct link between structural perturbations and the molecular pathogenesis of mitochondrial disorders. Furthermore, although NDUFA4 is expressed across most tissues, it has two tissue- and condition-specific isoforms thought to play adaptive roles, e.g., increasing CIV enzymatic efficiency and lowering the production of reactive oxygen species in upstream MRC enzymes[66]. NDUFA4L2 is induced by hypoxia[67] and highly expressed in the oxygen-sensing glomus cells in the carotid bodies[68], and NDUFA4L3/C15ORF48 is upregulated during inflammation[69,70] and

**Fig. 4 | Hierarchical incorporation of HIGD2A and NDUFA4 to CIV.**
**A**, **C**, **E**, **G** Steady-state levels of the indicated MRC complexes subunits assayed by SDS-PAGE and immunoblotting in mitochondria isolated from (A) WT, *HIGD2A*^KO, or *NDUFA4*^KO cells. **C** WT cells overexpressing HIGD2A (2A), NDUFA4 (A4), or carrying an empty vector (EV). **E** *HIGD2A*^KO cells overexpressing HIGD2A (2A), NDUFA4 (A4), or carrying an empty vector (EV). **G** NDUFA4KO cells overexpressing HIGD2A (2A), NDUFA4 (A4), or carrying an empty vector (EV). All experiments were repeated three times with similar results. **B**, **D**, **F** Distribution of HIGD2A and NDUFA4 in CIV and CIV-containing supercomplexes analyzed by BN-PAGE of isolated mitochondria from: (B) WT and *HIGD2A*^KO/*NDUFA4*^KO cells, solubilized with digitonin. The bar graph shows the quantification of the HIGD2A signal relative to WT cells in supercomplexes (SCs), complex IV (CIV), and the 50-kDa subcomplex IV (SubCIV),

presented as the mean ± SD from *n* = 5 independent experiments. Dots represent individual data points. Multiple two-sided unpaired *t*-tests were performed, and *p*-values were adjusted using the Benjamini-Hochberg FDR procedure. **p < 0.01, ***p < 0.0001. SCs: *p* = 6.8E-07, CIV: *p* = 0.000394, SubCIV: *p* = 0.0012. Exact test statistics are provided in the Source Data Table. **D** WT cells overexpressing HIGD2A (2A), NDUFA4 (A4), or carrying an empty vector (EV). **F** *HIGD2A*^KO cells overexpressing HIGD2A (2A), NDUFA4 (A4), or carrying an empty vector (EV). **H** *NDUFA4*^KO cells overexpressing HIGD2A (2A), NDUFA4 (A4), or carrying an empty vector (EV). Antibodies against ATP5A or VDAC1 were used as loading controls. IGA, in gel activity. All experiments were repeated three times with similar results. Source data are provided as a Source Data file.

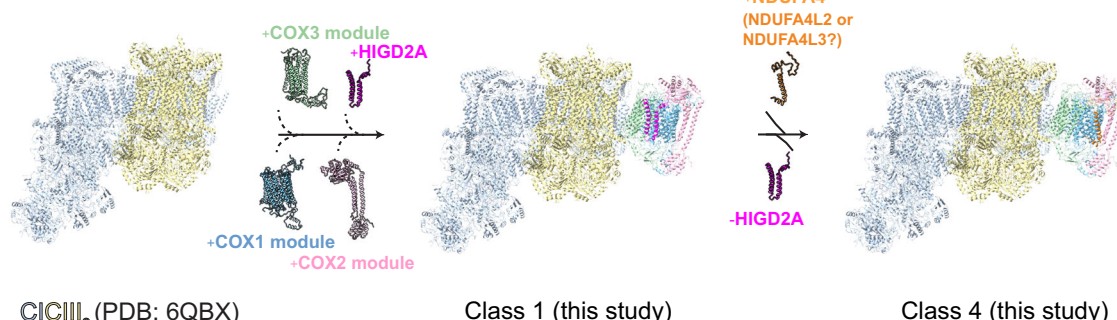

**Fig. 5 | Proposed model for the assembly of CIV within the respirasome CI + CIII$_2$ + CIV, based on the existing models and this study.** CI and CIII are shown as light steel blue and yellow, respectively. Multiple intermediate steps might exist between the CI + CIII$_2$ state and the class 1 (this study) state. The order of assembly of the COX modules into CI + CIII$_2$ has not been elucidated yet. The proteins from COX1, COX2, and COX3 modules are shown as light blue, light pink, and green, respectively. The HIGD2A and NDUFA4 (or alternatively NDUFA4L2 or NDUFA4L3) are shown as magenta and orange.

spermatogenesis[71]. Our findings show that a fraction of isolated and supercomplexed CIV is bound by HIGD2A. A hypothesis that warrants future investigations is that, in this way, HIGD2A may act as a placeholder, priming isolated and supercomplexed CIV for the rapid incorporation of different NDUFA4 isoforms, enabling cellular adaptation to stress.

## Method
### Human cell lines and cell culture conditions
Human HEK293T embryonic kidney cells (CRL-3216, RRID: CVCL-0063) were purchased from ATCC. A stable human *HIGD2A* knock-out (KO) line in the HEK293T background was previously constructed using the TALEN technology as described[23].

*NDUFA4*^KO human HEK293T cell pools were generated by Synthego Corporation (Redwood City, CA) by applying the CRISPR-Cas9 technology, using two guide RNAs targeted to exon 1 sequences in the *NDUFA4* locus (ACGGTAAGTGGCTGTAAATG and ACTTACGCTCG-GATGCTTCT). We used the pools for the clonal selection of *NDUFA4*^KO cell lines. Single clones were screened for NDUFA4 depletion by immunoblotting and genotyped by the PCR-amplification of the *NDUFA4* locus from the genomic DNA, using the primers SeqNDUFA4-F and SeqNDUFA4-R (Supplementary Table 5), followed by the sequencing of the resulting DNA fragments (Supplementary Fig. 9).

Cybrid cell lines were constructed using enucleated control fibroblasts and the osteosarcoma 143B TK 206 rho zero cell lines as described elsewhere[72]. *COX1* and *COX2* mutant cybrid cells carry a homoplasmic mitochondrial mutation; G6930A in the *COX1* gene, which generates a stop codon, and G7896A in the *COX2* gene, which generates a truncated version of the protein[60,61]. *COX3* mutant cybrid cell lines had a nearly homoplasmic frameshift mutation, 9537C$_{ins}$[62].

The cells were grown in high-glucose Dulbecco's modified Eagle's medium (DMEM, Life Technologies) supplemented with 10% (v/v) fetal

bovine serum (FBS), 2 mM L-glutamine, 1 mM sodium pyruvate, and 50 µg/ml uridine at 37 °C under 5% CO$_2$.

Stable mammalian cell lines that enable doxycycline-inducible expression of C-terminally FLAG-tagged COX14 (COX14::FLAG) were generated using the Flp-In T-Rex human embryonic kidney 293 (HEK293) cell line (Invitrogen). The COX14 cDNA was cloned into pcDNA5/FRT/TO (Supplementary Table 6). HEK293 cells were cultured in DMEM (Dulbecco's modified eagle medium) supplemented with 10% (v/v) FBS, 2 mM Glutamax (Gibco), and 1 × penicillin/Streptomycin (Gibco). 50 µg/ml uridine, 10 µg/ml Zeocin (Invitrogen), 100 µg/ml blasticidin (Gibco) at 37 °C under 5% CO$_2$ atmosphere. Before transfection, cells were seeded in a 6-well plate and cultured in culture media devoid of selective antibiotics. Cells were transfected with the pcDNA5/FRT/TO COX14::FLAG (Supplementary Table 6) using Lipofectamine 3000 (Invitrogen) following the manufacturer's recommendations. Fresh media supplemented with hygromycin (100 µg/ml, Invitrogen) and blasticidin (100 µg/ml) was added 48 h after transfection for positive clone selection. After two to three weeks, selected colonies were picked and cultured as single clonal populations. Cells were treated with 50 ng/ml doxycycline to confirm the inducible expression of COX14, and 24 or 48 h later, cell pellets were collected for western blot examination. The cell lines were routinely checked for mycoplasma contamination.

To rule out off-target effects during gene editing, all KO clones were reconstituted with the wild-type version of the corresponding gene. *HIGD2A*-Myc-DDK and *NDUFA4* were cloned under the control of a CMV promoter in the pCMV6-A-Entry-Hygro plasmid (Supplementary Table 6) using *Sfg*1-*Pme*1 or *Not*1 sites, respectively. For transfection of the different constructs, we used Lipofectamine™ 3000 Transfection Reagent (Invitrogen, Cat#L3000-015) according to the manufacturer's instructions. Media was supplemented with 200 mg/ml of hygromycin after 24 h, and drug selection was maintained for at least 21 days.

## Mitochondria isolation

Mitochondria-enriched fractions were isolated from 80% confluent 15 cm diameter plates as described[73] with the modifications stated below. Briefly, cells were collected by trypsinization and homogenized with ~20 strokes of a Potter-Elvehjem homogenizer in isolation buffer (20 mM HEPES-KOH, 220 mM Mannitol, 70 mM sucrose, 1 mM EDTA and 0.5 mM PMSF). Subsequently, differential centrifugation was performed to separate the mitochondria-enriched fraction. First, a low-speed centrifugation step ($800 \times g$ for 5 min) allowed the separation of intact cells, membranes and nuclei from the cytosol, and secondly, two higher speed centrifugation steps ($16000 \times g$ for 15 min) were made to isolate mitochondria. Finally, the mitochondrial pellet was resuspended in sucrose buffer (10 mM HEPES, pH 7.6 and 0.5 M sucrose). Protein concentration was determined by the Lowry method.

For cryoEM experiments, isolation of mitochondria was performed as previously described in Rorbach et al[74]. In brief, COX14::FLAG cell lines were induced with 50 ng/ml of doxycycline for 60 h and collected at a confluency of ~ 90% prior to mitochondrial isolation. The cell pellet was resuspended in the cold hypotonic MSE buffer (0.6 M mannitol, 10 mM Tris–HCl, pH 7.4, 1 mM EDTA, 0.1% BSA), and ruptured on ice by a semi-automatic homogenizer (Schuett-biotech). The lysate was clarified by centrifugation at $400 \times g$ and 4 °C for 10 min. The pellet was resuspended and subsequently homogenized. After 3 cycles of homogenization-centrifugation, the cell lysates were combined, and the mitochondria were pelleted by additional centrifugation at $11,000 \times g$ and 4 °C for 10 min. The crude mitochondria were loaded onto the sucrose cushion (1.0 M and 1.5 M sucrose in 20 mM Tris–HCl, pH 7.4, 1 mM EDTA) and centrifuged for 1 h at $77,000 \times g$ (25,000 rpm) in a SW41 Ti rotor (Beckman Colter). The band formed by the mitochondria in the middle between 1 and 1.5 M sucrose was collected carefully and resuspended in 10 mM Tris–HCl, pH 7.4 in a 1:1 ratio, The pure mitochondrial pellet was collected after centrifugation at $11,000 \times g$ and 4 °C for 15 min and then resuspended in mitochondrial freezing buffer (300 mM trehalose, 10 mM Tris–HCl pH 7.4, 10 mM KCl, 0.1% BSA, 1 mM EDTA), flash-frozen in liquid N2 and stored at −80 °C.

## SDS-PAGE and immunoblotting

To solubilize mitochondria in denaturing conditions, they were pelleted down and resuspended in 1X Laemmli buffer (2%SDS, 10% glycerol, 60 mM Tris-HCl, pH 6.8, 2.5% β-mercaptoethanol and bromophenol blue). 40–80 µg of mitochondrial protein extract was separated by sodium dodecyl sulfate-polyacrylamide gel electrophoresis (SDS-PAGE). Then, proteins were transferred to nitrocellulose membranes and incubated with specific primary antibodies listed in Supplementary Table 7. Peroxidase-conjugated anti-mouse and anti-rabbit IgGs were used as secondary antibodies (Molecular Probes). Optical densities of the immunoreactive bands were measured using the ImageJ software in digitalized X-ray film images.

## Blue native electrophoresis, 2-dimension electrophoresis and In gel activity assays

To extract mitochondrial proteins in native conditions, mitochondria were pelleted and solubilized in a buffer containing 1.5 M aminocaproic acid and 50 mM Bis-Tris (pH 7.0). After optimizing solubilization conditions, we decided to use digitonin with a detergent-to-protein ratio of 4 g/g. Solubilized samples were incubated on ice for 10 min and cleared at $20,000 \times g$ for 30 min. For gel loading, the supernatant was combined with sample buffer 10X (750 mM aminocaproic acid, 50 mM Bis-Tris, 0.5 mM EDTA and 5% Serva Blue G-250).

Blue Native electrophoresis was performed as described in detail elsewhere[75]. Briefly, Native PAGE Novex® 3%–12% Bis-Tris Protein Gels (Life Technologies) were loaded with 60–100 µg of mitochondrial protein sample. After electrophoresis, proteins were transferred to

PVDF membranes with the eBlot L1 Protein Transfer System (GenScript) and subjected to immunoblotting as indicated before.

In some of the BN gels, the bands corresponding to CIV or SC $CI + CIII_2 + CIV_n$ were excised from the gel and run on a second dimension SDS-PAGE after their immersion for 20 min in a denaturing solution (0.1% SDS and 0.07% β-mercaptoethanol) (Supplementary Fig. 8B).

Other gels were used for CIV activity determination by incubating them in 18 mL of phosphate buffer, pH 7.4, with 1.5 g sucrose, 10 mg 3.3'-diamidobenzidine tetrahydrochloride (DAB), 20 mg of cytochrome *c*, and 0.4 mg of catalase. Gel was incubated for up to 48 h at 37 °C in this solution. Pictures were taken at 4, 24 and 48 h of incubation.

## Immunoprecipitation

For the immunoprecipitation of HIGD2A-interacting proteins, mitochondria isolated from HEK293T cells overexpressing HIGD2A-Myc-DDK were solubilized as previously described. HIGD2A was pulled down using anti-Flag agarose beads (Sigma, Cat# A2220), and interacting proteins were eluted with 1% SDS. Similarly, NDUFA4 immunoprecipitation was performed using mitochondria isolated from wild-type HEK293T cells. An anti-NDUFA4 antibody (Supplementary Table 7) conjugated to protein A agarose beads (Thermo Scientific, Cat#20365) was used, and interacting proteins were eluted with 1% SDS.

## Measurement of mitochondrial respiratory chain complex IV activity and endogenous cell respiration

Mitochondrial cytochrome *c* oxidase (complex IV, CIV) activity was assessed spectrophotometrically as previously described[76]. Assays were performed using whole cell preparations from WT, *NDUFA4*-KO, and KO cells reconstituted with WT *NDUFA4*. Enzymatic activity was normalized to protein content and expressed relative to WT values.

Endogenous cellular respiration (CR) was measured polarographically at 37 °C using a Clark-type oxygen electrode (Hansatech Instruments, Norfolk, UK)[77]. Trypsinized cells were washed with cold PBS and resuspended at ~$3 \times 10^6$ cells/mL in 0.5 mL of respiration buffer containing 0.3 M mannitol, 10 mM KCl, 5 mM MgCl₂, 0.5 mM EDTA, 0.5 mM EGTA, 1 mg/mL BSA, 10 mM KH₃PO₄ (pH 7.4), and 2 mM ADP freshly added. The cell suspension was immediately transferred to the polarographic chamber for measurement of endogenous CR. To stimulate maximal cytochrome *c* reduction, 10 mM ascorbate and 0.2 mM N,N,N',N'-tetramethyl-p-phenylenediamine (TMPD) were added in a set of experiments. In a separate series, endogenous coupled respiration was uncoupled by incremental additions of up to 0.4 µM carbonyl cyanide p-trifluoro-methoxyphenyl hydrazone (FCCP) to determine the maximal oxygen consumption rate. In both sets of experiments, CIV-dependent respiration was confirmed by inhibition with 0.8 µM potassium cyanide (KCN). All respiration rates were normalized to total cell number.

## Sample preparation for cryoEM

Isolation and purification of mitochondria from a large-scale COX14::FLAG overexpressing cell line were performed as described in the above paragraph "Isolation of mitochondria". The pure mitochondria were thawed slowly on ice, then lysed at 4 °C for 40 min in a lysis buffer (25 mM HEPES-KOH, pH = 7.5, 50 mM KCl, 20 mM Mg(OAc)₂, 1% (v/v) LMNG, 0.1 % β-DDM, 0.01 % cardiolipin, 0.1 mM DTT, 1x cOmplete EDTA-free protease inhibitor cocktail (Roche), 40 U/µL RNAse inhibitor (Invitrogen)). The mitochondrial lysate was centrifuged at $20,000 \times g$ for 15 min at 4 °C. The supernatant was then added to ANTI-FLAG M2-Agarose Affinity Gel (Sigma-Aldrich), previously equilibrated with the lysis buffer and incubated for 2 h at 4 °C. After incubation, the mixture was centrifuged at $5000 \times g$ for 1 min at 4 °C, the supernatant was removed, and the gel was washed three times with the wash buffers (25 mM HEPES-KOH, pH = 7.5, 50 mM KCl,

20 mM Mg(OAc)$_2$, 0.1% (v/v) LMNG (Thermo Fisher), 0.01 % cardiolipin, 0.1 mM DTT). Elution was performed twice by adding the elution buffer (25 mM HEPES-KOH, pH = 7.5, 50 mM KCl, 20 mM Mg(OAc)$_2$, 0.01% (v/v) Lauryl Maltose Neopentyl Glycol (LMNG), 0.001 % cardiolipin, 0.001 % glycol-diosgenin (GDN, Sigma-Aldrich), 0.1 mM DTT, 500 µg/mL 3X FLAG peptide (Sigma-Aldrich) for about 45 min at 4 °C. The elution was combined, then centrifuged at 55,000 rpm in TLA55 rotor (186,000 × g) for 16 h. Most of the supernatant was discarded, and the last 20 µL at the bottom was collected. Fifteen µl of elution buffer without FLAG peptide was added to resuspend the pellet by slightly pipetting. The tube was centrifuged at 20,000 × g for 10 min to get rid of any insoluble part, the supernatant was then combined with 20 µL above, centrifuged one more time at 20,000 × g for the last 15 min and used directly to prepare the grid. Grids were prepared using protein at a concentration of 1 mg/ml.

## Cryo-EM data acquisition and data processing

Prior to cryo-EM grid preparation, grids were glow-discharged with 25 mA for 120 s using an EMS100X easiGlow glow-discharge unit. Quantifoil Cu 300 mesh (R 2/2 geometry – Agar Scientific) grids covered with a thin carbon layer of ~ 2 nm were used for the preparation of the cryo-EM samples. Four µl of the sample was applied to the grids, incubated for 30 s, then vitrified at 4 °C and 100 % humidity using a Vitrobot Mk IV (Thermo Fisher Scientific) (blot 5 s, blot force 3, 595 filter paper).

All cryo-EM data collection (Supplementary Table 1) was performed with EPU (Thermo Fisher Scientific) operated on Titan Krios G3i transmission electron microscope (Thermo Fisher Scientific) at 300 kV in the Karolinska Institutet's 3D-EM facility. Images were acquired in nanoprobe 105kX EFTEM SA mode (0.84 Å/px) with a slit width of 20 eV using a K3 Bioquantum, for 3.5 s with 50 movie frames and a total electron dose of 35 e-/Å$^2$. Images were collected with a targeted defocus range of −0.6 to −2 µm, with an autofocus routine at every acquisition area. A total of 17,476 movies were collected from 2 data sets.

Data processing was performed using cryoSPARC (v3.2–v4.2)[78]. Movie frames were aligned using Patch Motion Correction, and Contrast Transfer Function (CTF) parameters were estimated by Patch CTF correction. Particle picking was performed by automatic Gaussian blob detection (mask diameter 250–500 Å with circular blob), yielding particles that were then subjected to reference-free 2D classifications (number of classes = 200). A total of 2,508,182 particles were initially picked from two data sets. Particles were inspected, then extracted in a box size of 600 Å, and Fourier cropped to a box size of 200 Å. Iterative 2D classifications were used to select the best 2D class averages of respirasome-like particles. The ab initio reconstruction was used to create four classes. Several rounds of heterogeneous refinement were applied to obtain one of the four classes, yielding a respirasome reconstruction with high-resolution features that contains 375,422 particles. The particles were re-extracted to the full box size of 600 Å for further refinement. After homogeneous refinement of these particles, 3D classification without mask followed by iterative heterogeneous refinements was performed to get rid of classes of particles with broken densities, resulting in 336,847 particles yielding 3.1 Å resolution after homogeneous refinement (Supplementary Fig. 1). These particles were utilized for the subsequent refinement of individual complexes.

Masked refinements of the respective SC intermediates resulted in map resolutions of 2.9 Å for CI, 2.8 Å for CIII, and 3.1 Å for CIV. For CI and CIII refinement, the 3D classification using the focus mask (Supplementary Fig. 1) further discarded the subsets of particles displaying minor populations, resulting in 236,346 particles and 213,731 particles, respectively, for the final refinement. It should be noted that the unused particles exhibited lower resolution structural features or appeared to represent minor conformational states. These were not further analyzed in this study to focus on predominant structural states, as the conformational change analysis of CI and CIII has been intensively studied previously[40,42–44,79,80]. The resulting particles were subjected to global CTF refinement and local CTF refinement, yielding the final maps with 2.66 Å and 2.52 Å resolution (FCS 0.143) for CI and CIII, respectively (Supplementary Fig. 1, Supplementary Table 1).

For complex IV, a mask covering the COX3 module (COX3, COX5B, COX6A1 and COX7A2) and regions of NDUFA4 and HIGD2A was created. The 3D classification using this mask was performed with an output of 8 classes. These classes were investigated and combined into 4 classes representing different states (Fig. 1B, Supplementary Fig. 1). They include two major classes: class 1 contains clear extra density for HIGD2A protein, while class 4 contains clear density for NDUFA4 protein. Two other classes, class 2 and class 3, show continuous densities for both proteins, representing the intermediates. Class 2 expresses clear density for HIGD2A protein and incoming density of NDUF4A, class 3 shows clear density for NDUFA4 and weak density at HIGD2A binding region (Fig. 1B). These particles were also subjected to global and local CTF refinement, followed by local refinement, yielding the maps with a range from 2.9 to 3.6 Å resolution (Supplementary Figs. 1, 2 Supplementary Table 1). The final maps of CI, CIII, class 1 and class 4 of CIV from local refinement jobs were subjected to an autosharpen job in Phenix v1.21.1[81], and the output maps were used for model building.

## Model building and refinement

The atomic models were built manually in COOT[82]. First, the previous structures of human CI (PDB: 5XTD)[14], human CIII$_2$ (PDB: 5XTE)[14] and human CIV (PDB: 5Z62)[54] were fitted into the corresponding map as a rigid body using Chimera X v1.6[83,84]. Then, the fitted model was manually adjusted in COOT in a residue-by-residue manner against the density map using Ramachandran restraints, torsion restraints and appropriate map weight. Densities for lipids were manually fitted by using previously reported structures as references (PDB: 8OM1[47] and 5XTD[14] for CI; 5XTE[14] and 8UGD[10] for CIII; 5Z62[54] and 8UGL[10] for CIV). The ligands, cofactors and modifications of proteins or subunits in individual complexes were manually inspected according to the Uniprot database and experimentally determined structures reported previously (PDB: 8OM1 for CI; 8UGD for CIII; and 5Z62 for CIV). The sequence of the HIGD2A protein was built in the extra density region of class 1 of CIV with the reference model (AF-Q9BW72-F1) from the AlphaFold protein structure database[85]. Hydrogens were added to the models of CI and HIGD2A-bound CIV in Refmac5[86] of the CCP4 suite[87] and to the model of CIII with Phenix.readyset[88]. The model was further real-space refined in Phenix v1.21.1[81] by subjecting it to 4–6 cycles of global energy minimization, local grid search and atomic displacement parameter (ADP) refinement. This was done using reference model restraints, with the model itself used as the reference. Additionally, Ramachandran restraints set to Oldfield (favored) and Emsley8k (allowed and outlier); and rotamer restraints targeting outliers (to fit "outliers and poor map") were applied. Restraints for ligands were generated with Refmac5[86] and used for model refinement in Phenix v1.21.1[81].

## Quantification and statistical analysis

Unless indicated, all experiments were performed at least in biological triplicate, and the results were presented as mean ± standard deviation (SD) of absolute values relative to control groups. Statistical p-values for comparison of two groups were obtained by application of a Student's two-tailed unpaired t-test, and a p < 0.05 was considered significant.

## Reporting summary

Further information on research design is available in the Nature Portfolio Reporting Summary linked to this article.

## Data availability

All cryo-EM maps and atomic coordinates have been deposited at the Electron Microscopy Data Bank (EMDB) and the Protein Data Bank (PDB) as follows: EMD-52596 (initial consensus map); complex I: EMD-54784/PDB 9TI4; complex III EMD-52525/PDB 9HZL; complex IV: EMD-52664 (initial local refinement of consensus map), different states of complex IV: EMD-52654/PDB 9I6F (class 1: HIGD2A bound complex IV), EMD-52612 (class 2: complex IV with HIGD2A density and incoming NDUFA4), EMD-52613 (class 3: NDUFA4 bound complex IV with weak density from HIGD2A), EMD-52662/PDB 9I7U (class 4: NDUFA bound complex IV). Source data are provided with this paper.

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

## Acknowledgments

We thank A. Bondy and the Karolinska Institutet's 3D-EM facility and the cryo-EM Swedish National Facility (funded by KAW, EPS and Kempe foundations) for assisting during data collection. We thank R. Baradaran for helping with the initial stage of this project, H. Das and M. Hallberg for assisting during data processing and model building. This research was supported by Karolinska Institute, the Knut & Alice Wallenberg Foundation (WAF2017 and KAW 2018.0080 to J.R.), the Swedish Research Council (VR2016-02179 to J.R.), National Institute of General Medicine (NIGMS) [R35-GM118141 to A.B.]. M.D.N. was supported by an EMBO postdoctoral fellowship (LTF-2020-606). A.S-M. was supported by The American Heart Association (AHA) postdoctoral fellowship (25POST1374467, doi: [https://doi.org/10.58275/AHA.25POST1374467.pc.gr.227457]). A.B. is the recipient of a BLRD Research Career Scientist Award IK6BX006815. Some figures were created using BioRender (BioRender.com) and are mentioned in figure legends.

## Author contributions

Methodology: M.D.N., A.S-M., and A.T-G. Analyses: M.D.N., A.S-M, A.K, V.S, A.B, and J.R. Visualization: M.D.N., A.K., and V.S. Writing Original Draft: M.D.N., A.S-M. A.B., and J.R. Review & Editing: all authors. Funding Acquisition: A.B. and J.R.

## Funding

## Competing interests

The authors declare no competing interests.
