## [Transparent Peer Review file · Nature Communications]

Structural basis for late maturation steps of mitochondrial respiratory chain complex IV within the human respirasome

Corresponding Author: Dr Joanna Rorbach

Version 0:

Reviewer comments:

Reviewer #1

(Remarks to the Author)

Summary

In this manuscript, the authors use cryo-EM to solve structures of the human respirasome, revealing a mixture of a fully assembled supercomplex bound to NDUFA4 and one featuring an immature CIV bound to HIGD2A. The structures reveal that HIGD2A partially occupies the position of CIV subunit NDUFA4, concluding that HIGD2A acts as a placeholder for NDUFA4. Complementary biochemical analysis shows that NDUFA4 assembles late into CIV and in absence of NDUFA4, HIG2DA binds more stably to HIG2DA, supporting the role of HIGD2A as placeholder for the assembly of the late-stage CIV subunit NDUFA4. The manuscript is relevant to the field as it adds an important piece of the puzzle with regards to the assembly mechanism of supercomplexes: it is in fact the first structural evidence in support of the integrated assembly mechanism of respirasomes, where intermediates of complexes come together in the respirasome as opposed to fully assembled complexes forming the respirasome.

I believe this manuscript is of interest for a broad audience due to the implications for human metabolism and disease, however I think the authors need to address some issues before the paper can be published.

My main concerns are:

1) Some data is missing, such as:

a) particle numbers indicated in the processing pipeline don't add up and there is no explanation for what was discarded and why

b) the local resolution and angular distribution plots for the final cryo-EM maps have not been reported

c) rationale for COX14-FLAG purification when COX14 does not appear to be part of the structures is not explained

2) The conclusions made based on the provided maps are overstated:

a) the resolution is not high enough in TMH2 of COX2 (discussed in relation to Fig 3, regarding the K-pathway) discuss conformational changes at the side-chain level

b) the side-chain densities for NDUFA4 and HIGD2A are only visible for some residues, not allowing for exclusive density-based subunit assignment.

The main problem in this regard seems to be that the overall resolution indicated in the text and in Fig S1 is way better than the local resolution effectively reached in the above-mentioned regions of the structure. The issue would probably be apparent from the local resolution plots, but these are not featured in the current manuscript. It is therefore crucial that the authors provide the missing information and revise the results section to avoid overinterpreting the data.

3) The functional relevance of HIGD2A as a placeholder for NDUFA4 is unclear.

More details on my main concerns, as well as some minor comments, are included in the text below.

Detailed comments

In line 36, "SC assembly" should be replaced by "the canonical respirasome assembly", as the findings of the paper only pertain to this supercomplex.

Lines 40 to 43: unclear. Which binding factors are the authors referring to here? None seems to be mentioned in the manuscript.

Line 54: the reducing equivalents that fuel the MRC are contained in NADH and succinate, not NADH and FADH2. FADH2 is a cofactor of CII, analogous to FMN in CI, while NADH and succinate are the reduced species that donate electrons to CI and CII, respectively.

Lines 128-129: where is COX14 in the structures? Since it is used for affinity purification, it would be expected to find it in the structures of the purified material. It appears that COX14 is an assembly factor involved in the biogenesis of the COX1 module, so it is not clear how it would be bound to/allow purification of late assembly intermediates and fully assembled

supercomplexes. The authors should clarify this aspect. It is possible that the observed structures derive from the extremely abundant respirasomes present in the mitochondrial membranes, but this would mean the structures come from “contaminants” of the purification and this should be stated clearly. The current wording is misleading, as it suggests that tagging COX14 was necessary to purify the solved structures. If instead this was indeed the case, the authors should explain how COX14 would allow the purification of fully assembled and late assembly respirasomes.

Lines 139-140: where are the results of 3D variability analysis of CIV shown?

Lines 248-251: based on the provided map (HIGD2A_bound_complexIV_map.ccp4), it does not appear that most side chains from residue 31 to 106 show unambiguous density. As commented below for figure S1, it appears that the reported resolutions don't match the quality of the maps and the peripheral region of CIV, where HIGD2A is located, is strongly affected by the issue. While there is density for some bulky residues (such as Phe39, Tyr62, Arg80) and the overall topology is in good agreement with the alphafold prediction provided by UniProt

(<https://www.uniprot.org/uniprotkb/Q9BW72/entry#structure>), which point towards the density being indeed HIGD2A, “with sufficient resolution to unambiguously resolve most amino acid side chains from Asn31 to Pro106 at the distal C-terminal domain” is an overstatement.

Lines, 302-304: as above, the resolution of the map region corresponding to NDUFA4 appears to be worse than 3.2 Å. Although the agreement with the previously reported structure (PDB 5Z62) and the good side chain density for some bulky residues (such as Phe19 or Tyr 32) support the density assignment, which is hereby not disputed, the wording in the text is misleading. I suggest removing this sentence.

Lines 258-270: it would be good to add to Fig S5 a comparison of the CIV structures from yeast CIII2CIV/CIII2CIV2 featuring Rcf and the human CIV featuring HIGD2A from this study, to make it easier to follow the paragraph. Also, in line 263 the authors refer to Rcf1, while in the rest of the paragraph they refer to Rcf2, so Rcf1 appears to be a mistake.

Lines 332-334: based on the provided maps, the whole TMH2 of COX2 does not show very high-resolution density, therefore making side chain-based conclusions (showed in Fig 3C and S4 F-G) appears to be an overstatement. What appears to be data-supported is that NDUFA4 stabilizes TMH2 of COX2 compared to the HIGD2A-featuring CIV, but if the rationale provided in lines 327-328 is followed and higher flexibility is necessary for proton uptake, it is then unclear why an assembly intermediate (CIV featuring HIGD2A) would be more “proton uptake ready” than the fully assembled complex (CIV featuring NDUFA4). Related to this, the discussion (lines 459-463) should be edited accordingly.

Line 338: a comment remained in the text

Lines 338-339: Ref 53 already showed that NDUFA4 is a subunit of CIV, so while it is true that this study confirms the fact, I suggest the previous paper be cited here.

Line 340: the authors do not provide any evidence, apart from the structure-based speculation, that NDUFA4 regulates proton transfer by stabilizing the entry point of the K-pathway. I suggest rephrasing to “NDUFA4 binding might regulate proton transfer”, or providing biochemical evidence and/or references from other studies that this is indeed the case.

Lines 315-341: based on my comments on lines 332-334 and 340, I suggest the authors revise this section, alongside Figs 3 and S4, by (1) limiting the discussion to secondary structure-level conformational changes of COX2 TMH2, which are well supported by the data and (2) clarifying the implications of a NDUFA4-mediated stabilization of COX2 TMH2 for the proton transfer.

Figure 4D, there is no signal for HIGD2A in the WT in the respirasome area of the gel, making it difficult to conclude that overexpression of NDUFA4 reduces HIGD2A binding to CIV (lines 393-397), with relation to the respirasome assembly cited in line 404. Also, in Figure 4F there seems to be less NDUFA4 in the respirasome of HIGD2A-KO+NDUFA4 than in HIGD2A-KO+EV, but looking at the level of NDUFA4 in monomeric CIV or the ATP5A loading control it does not seem like the difference would be ascribed to loading different amount of material: what would be the explanation for this?

Lines 417-423: there are no bar graphs in the figure, but indeed it would be good to see them.

Lines 459-463: can the authors point to activity measurements proving that in absence of NDUFA4 oxygen reduction is less efficient?

Lines 464-466: how is the data helping understand the underlying pathogenic mechanisms of Leigh syndrome? What kind of information are the current structures adding to the state of the art? It appears that references 68 and 69 concern patients in which NDUFA4 is absent, or almost absent, so how are the presented data helping to explain the pathogenic mechanisms of Leigh syndrome?

Figure S1: The particle numbers don't add up. The consensus refinement indicates 301k particles, from which the local refinements are performed. But the total of the four CIV classes is around 334k particles, so it is not clear where the additional particles come from.

Additionally, it appears that only a subset of the particles were used to refine CI (168k particles) and CIII (213k particles). In the Methods section, the authors state that “the focus mask further removed the poor alignment particles”, but this does not seem very likely because in the respirasome CI and CIII usually drive the alignment, partly because the membrane arm of CI and the proximal protomer of CIII form the “rigid core” of the respirasome and also because the eight FeS clusters of CI and the six hemes of CIII provide strong densities that aid the particle alignment. It is therefore strange that CI and CIII would be poorly aligned in the respirasome, particularly CI which “loses” almost half of the particles from the consensus refinement to the focused refinement. As there is no data shown for the processing steps leading to the final maps of CI and CIII, one cannot see how the removed classes look like, or at which stages they were removed. It is therefore important that the authors show exactly how the data was processed for CI and CIII. Since so many particles are removed, is it possible that other intermediates, perhaps lacking the N-module of CI, are present in the dataset and were discarded as “poorly aligned”? Or are the discarded particles corresponding to the open state of CI? Also, what is the overlap between discarded particles for CI and CIII? Meaning, are the discarded particles coming from the same respirasome particles? If so, is it possible that those are generally damaged particles, which therefore should not be considered for CIV refinement either? When focused refinements are run starting from a consensus map, to boost the resolution, one would expect that the number of particles does not change because this is the pool that corresponds to the structure of interest, in this case the respirasome. Removing one third (as was done for CIII) or half of the particles (as was done for CI) is unexpected and appears unjustified, therefore needs to be clarified and supported by data.

Lines 676-678: Along the same line, in the methods section it is stated that the CIV focused classification was done with 6 classes, then the 6 classes should be shown in the supplementary figure, also indicating which classes were pooled based on their overlapping features.

Figure S1 lacks:

- 1) the map-to-model FSC curves for each modelled structure
- 2) the half-map FSC curve for the consensus map
- 3) the local resolution depictions (i.e. surface colored by local resolution), as well as
- 4) the angular distribution plots for each focused map and the consensus map.

The description of how the plots and colored maps are obtained should then also be added to the methods section. The methods section should also state whether the maps were sharpened and if so how.

Points 3 and 4 are particularly important because the provided maps don't look like they correspond to the reported resolutions, so the local resolution coloring would clarify which regions of the maps really reach the reported resolution and the angular distribution would shed light on a potential anisotropy issue, which might contribute to explaining why the maps don't look like they correspond to the reported resolution.

As the "class 2" and "class 3" complex IV maps provided don't look anisotropic, it might also be that the weird appearance of the ccp4 maps is a sharpening artifact, which is why it would be important to declare whether the maps were sharpened.

Minor comments

Figure S4: in panels F and G, the subunit to which Y244 and K319 belong (COX1 presumably) should be indicated in the legend.

All figures displaying densities (Figs. 1, S2, S3, S4) should indicate the contour used in the legend.

Line 604: the section is titled "immunoprecipitation", but lines 606-628 feature the description of sample purification for cryo-EM. I suggest this section is thus separated and titled "cryo-EM sample preparation"

Legend to Fig S2: state that NDP stands for NADPH, as this is not clarified.

Fig 2D: the figure, or legend, lack reference to the scalebar used to calculate the electrostatic potential.

Reviewer #2

(Remarks to the Author)

The assembly of the respiratory chain and supercomplex formation plays important roles in human health and disease. This paper describes a cryo-EM study of human respirasomes, purified using an engineered tag on a complex IV subunit. The analysis revealed I2III2IV1 supercomplexes with high-resolution structures of fully assembled CI and CIII, while CIV density is weak. Local refinement and classification of the flexibly attached CIV yielded distinct classes, containing either the assembly factor HIGD2A and/or the subunit NDUFA4, which is the last subunit to be added to the complex.

In this interesting paper the authors make a convincing case that the removal of HIGD2A is necessary for insertion of NDUFA4, as their positions partially overlap. However, the authors do not mention what, if any, role HIGD2A plays in respirasome formation. Also, it is not clear if acting as a "placeholder" for NDUFA4 is the only function of HIGD2A or if it plays a role in the attachment of the COX3 module, as previously suggested. The paper would be strengthened if these issues were addressed.

Specific comments:

1. The protein was purified with a tag on CIV, so presumably yielding both respirasomes and single CIV. However, the latter is never mentioned. Are the particles visible in the micrographs? If so, why are these not analysed? If not, how can this be explained?
2. The introduction, p. 4 states "it has been proposed that HIGD2A promotes the formation of respirasome assembly intermediates by coordinating the association of CIV assembly modules, revealing divergent assembly pathways for individual and supercomplexed CIV". Is there any evidence for this? For a distinction it would be necessary to study individual CIV as well. It is nowhere shown how CIV is oriented in the respirasome and what subunits interact with CI or CIII.
3. Further to that, the model in Fig. 5 does not show what induces CIV attachment to the respirasome.
4. p. 12: The story about the flexibility at the K-pathway entrance is confusing. The region is said to typically have high B factors, indicating flexibility, nevertheless distinct conformations are assigned to these uncertain densities. Particularly suspicious is Glu62: a negatively charged side chain typically has no cryo-EM density (unless the charge is compensated by a nearby positive charge like a salt bridge or ion ligand) and the side chain can't be modeled correctly even in a rigid structure. The absence of the glutamate sidechain is obvious in Suppl. Fig. 4F and G (although just showing density for one residue does not give an indication of the level of flexibility in this region). Thus, the suggestion that the conformation of Glu62 in a bovine CIV structure in GDN represents an intermediate state between the two human structures is a very weak argument (Fig. 3C). If the authors feel that there is a real conformational change in this region, better figures will be needed showing the actual densities, not just models. Otherwise, this section needs to be toned down. Further to suppl. fig. 4F,G: please indicate which subunits the highlighted residues belong to. Glu62 is misspelled as Glu61 in the legend.
5. Figure 1B: please indicate how CIV is oriented relative to the respirasome shown in 1A.
6. Cryo-EM methods (p.23):
 - a. What is the protein concentration?
 - b. What microscope was used?

- c. Please add “total electron dose” of 35 e-/A2.
- d. How many particles were picked?
- e. “bad alignment particles” and “poor alignment particles”: please rephrase.
- f. What is a “local refinement volume”?
- g. Global or local CTF refinement?

7. Model building (p. 24):

- a. Densities for lipids were manually fitted using references: please explain.
- b. “The ligands, cofactors and modifications were manually inspected as suggested in Uniprot and experimentally determined structures reported previously”. Unclear.
- c. HIGD2A was manually built with an AlphaFold structure as reference. Please remove “manually”.

8. Suppl. fig. 1: The workflow needs more details/images: exemplary micrographs, 2D class averages, number of particles picked initially, initial models.

Version 1:

Reviewer comments:

Reviewer #1

(Remarks to the Author)

I thank the authors for thoroughly considering and addressing my comments. I am satisfied with the revised version of the manuscript and recommend publication without further revision.

Reviewer #2

(Remarks to the Author)

The authors have properly addressed most of my comments. One issue remains: the interpretation of differences in the K pathway between complex IV maps in the presence of NDUFA4 or HIGD2A. Line 360: “The density for Glu62 of COX2 in the HIGD2A-bound CIV suggests uncertainty regarding side chain conformation, consistent with high flexibility (Supplementary Figure 5F). Interestingly, this region appears slightly more ordered in the presence of NDUFA4 (Supplementary Figure 5G), indicating a potential stabilizing effect.”. In the revised manuscript the authors have added local resolution plots, which clearly show that the resolution in this region is poor in both maps. But in fact, looking at the maps provided by the authors, both CIV maps also suffer from anisotropic resolution, which is not visible from a static picture. Due to all these factors, even the backbones can't be reliably interpreted in either map, let alone sidechains. And, as stated in my original review, negatively charged sidechains typically have no cryo-EM density. So the “uncertainty regarding side chain conformation” is not even due to the low resolution.

Further, the authors state that “this region appears slightly more ordered in the presence of NDUFA4 (Supplementary Figure 5G), indicating a potential stabilizing effect”. Sfig. 5 F and G show density for Glu62 only, in 5G at a very low contour level where there is a blob of density assigned to the sidechain. The figure is missing all the context of the maps, where there is no indication of “better ordering” in one of them.

Figure 3C shows the “conformational change” of Glu62, including a previously published structure (8D4T from EMD-27196). I checked that map and although it has better resolution than the ones from the present study, there is no density for the (charged) Glu62 sidechain and its conformation in the structure is based on guesswork.

In short, the authors do not provide any evidence for a conformational change or lower flexibility due to NDUFA4 binding. The region is simply flexible in all structures. It would be best if the whole story (line 340-364) is eliminated, including Figure 3C and Sfig. 5F and 5G.

Response to reviewers' comments

We thank the reviewers for their time and effort in evaluating our manuscript and for raising insightful questions. Their detailed and thoughtful feedback has been instrumental in enhancing the quality of our work.

Reviewer #1 (Remarks to the Author):

Summary

In this manuscript, the authors use cryo-EM to solve structures of the human respirasome, revealing a mixture of a fully assembled supercomplex bound to NDUFA4 and one featuring an immature CIV bound to HIGD2A. The structures reveal that HIGD2A partially occupies the position of CIV subunit NDUFA4, concluding that HIGD2A acts as a placeholder for NDUFA4. Complementary biochemical analysis shows that NDUFA4 assembles late into CIV and in absence of NDUFA4, HIG2DA binds more stably to HIG2DA, supporting the role of HIGD2A as placeholder for the assembly of the late-stage CIV subunit NDUFA4. The manuscript is relevant to the field as it adds an important piece of the puzzle with regards to the assembly mechanism of supercomplexes: it is in fact the first structural evidence in support of the integrated assembly mechanism of respirasomes, where intermediates of complexes come together in the respirasome as opposed to fully assembled complexes forming the respirasome.

I believe this manuscript is of interest for a broad audience due to the implications for human metabolism and disease, however I think the authors need to address some issues before the paper can be published.

We would like to thank the reviewer for their positive evaluation and thorough analysis of the manuscript. This careful set of comments is greatly appreciated.

My main concerns are:

1) Some data is missing, such as:

a) particle numbers indicated in the processing pipeline don't add up and there is no explanation for what was discarded and why

We have revised Supplementary Figure 1 to provide additional details on data processing, including updated particle counts and an explicit indication of discarded particles. Corresponding updates have also been made to the Data Processing section of the Methods.

Lines 732-755:

A total of 2,508,182 particles were initially picked from two data sets. Particles were inspected, then extracted in a box size of 600 Å and Fourier cropped to a box size of 200 Å. Iterative 2D classifications were used to select the best 2D class averages of respirasome-like particles. The ab initio reconstruction was used to create four classes. Several rounds of heterogeneous refinement were applied to obtain one of the four classes, yielding a respirasome reconstruction with high resolution features (375,422 particles). The particles were re-extracted to the full box size of 600 Å for further refinement. After homogeneous refinement of these particles, 3D classification without mask followed by iterative heterogeneous refinements was performed to get rid of classes of particles with broken densities, resulting in 336,847 particles yielding 3.1 Å resolution after homogeneous refinement (Supplementary Figure 1). These particles were utilized for the subsequent refinement of individual complexes.

Masked refinements of the respective SC intermediates resulted in map resolutions of 2.9 Å for CI, 2.8 Å for CIII, and 3.1 Å for CIV. For CI and CIII refinement, the 3D classification

using the focus mask (Supplementary Figure 1) further discarded the subsets of particles displaying minor populations, resulting in 236,346 particles and 213,731 particles, respectively, for the final refinement. It should be noted that the unused particles exhibited lower resolution structural features or appeared to represent minor conformational states. These were not further analyzed in this study to focus on predominant structural states, as the conformational change analysis of CI and CIII has been intensively studied previously¹⁻⁶. The resulting particles were subjected to global CTF refinement and local CTF refinement, yielding the final maps with 2.66 Å and 2.52 Å resolution (FCS 0.143) for CI and CIII, respectively.

b) the local resolution and angular distribution plots for the final cryo-EM maps have not been reported

We have added a new Supplementary Figure 2, which includes local resolution and angular distribution for all reported maps.

c) rationale for COX14-FLAG purification when COX14 does not appear to be part of the structures is not explained

We employed COX14-FLAG purification to capture assembly intermediates of CIV (as described previously by Richter-Dennerlein et al, doi: 10.1016/j.cell.2016.09.003). Unexpectedly, the most prominent and stable complexes obtained through our procedure, and subsequently visualized by cryo-EM, were supercomplexes, whereas COX14 itself was not detected in the final structures. Notably, our immunoprecipitation analyses, together with a closer examination of the mass spectrometry data reported by Richter-Dennerlein et al., support the presence of the supercomplexes interacting with COX14. As this finding has been observed across different laboratories, it is unlikely to represent an artifact of our procedure. Instead, it suggests that the insertion of mitochondrially encoded proteins into CIV occurs in the context of supercomplexes—an observation we emphasize throughout the manuscript.

Reciprocal IP analyses using HIGD2A and NDUFA4 as baits (Figure 1) also detected COX14, although only a small fraction of COX14 was recovered in the eluates, suggesting a transient interaction with the captured complexes. In our experience, it is common for bait proteins to be partially or fully lost during purification, particularly when the interaction is weak or transient. Similar challenges have been reported by others: for example, Lenaric et al. <https://doi.org/10.1038/s41467-021-23811-8> used MTG1-FLAG to purify the mitoribosome but did not visualize MTG1 in any observed mitoribosomal large subunit assembly states, while Cheng et al. <https://doi.org/10.1038/s41467-021-24818-x> reported that GTPBP10 was present in only ~1% of mitoribosomal complexes despite being used as bait.

In the case of our COX14 IP, additional post-immunoprecipitation centrifugation steps likely contributed to its loss. Specifically, we performed a 16-hour centrifugation at 55,000 rpm in a TLA55 rotor to enrich for higher-molecular-weight complexes and minimize interference from smaller complexes in downstream cryo-EM analyses (see Sample preparation for cryo-EM in the Methods section).

As the mechanistic basis of COX14 interactions turned out not to be the primary focus of this study, it remains to be explored in future work. Nonetheless, we have clarified in the main text that COX14 was not detected in the structural analysis, likely due to its transient interaction and subsequent loss during post-IP preparations.

We added this sentence to clarify in the manuscript

Line 130-132:

To gain insight into the assembly of CIV, we engineered a C-terminal FLAG tag on the CIV biogenesis factor COX14, which has previously been shown to associate with COX1 during its translation and CIV maturation within the IMM^{7,8}.

Line 140-143:

Despite extensive data processing and map analysis, we were unable to identify any additional density corresponding to COX14, the bait protein, most likely due to its transient interaction and subsequent loss during post-immunoprecipitation processing.

2) The conclusions made based on the provided maps are overstated:
a) the resolution is not high enough in TMH2 of COX2 (discussed in relation to Fig 3, regarding the K-pathway) discuss conformational changes at the side-chain level
b) the side-chain densities for NDUFA4 and HIGD2A are only visible for some residues, not allowing for exclusive density-based subunit assignment.

The main problem in this regard seems to be that the overall resolution indicated in the text and in Fig S1 is way better than the local resolution effectively reached in the above-mentioned regions of the structure. The issue would probably be apparent from the local resolution plots, but these are not featured in the current manuscript. It is therefore crucial that the authors provide the missing information and revise the results section to avoid overinterpreting the data.

Thank you for pointing this out. We agree that the local resolution in peripheral regions, including NDUFA4 and HIGD2A, is lower than the overall resolution indicated in the text and Supplementary Figure 1. We have now included maps showing local resolution in Supplementary Figure 2 and revised the text accordingly. In addition, we have removed the word ‘unambiguous.’

Line 257-260:

An additional density corresponding to a polypeptide with 2 transmembrane helices was observed near COX3 and COX6A, but at a relatively lower local resolution in the range of 3.5 to 6 Å (Supplementary Figure 2, class 1 of complex IV).

Although the resolution of TMH2 of COX2 is also lower than GSFSC resolution, the electron density for Glu62 residue in NDUFA4-bound complex IV is better visualized than that in HIGD2A-bound complex, as shown in Supplementary Figure 5F and 5G (Supplementary Figure 4 in the previous version). Furthermore, we added Supplementary Figure 5H and 5I to display the B factor of the COX2 subunit in both structures. While the B-factor ranges are almost similar for the two models, the region of the K-pathway entrance (Glu62 as a representative residue) in the HIGD2A-bound state displays a greater B-factor than that in the NDUFA4-bound state. Therefore, we concluded that the region in the HIGD2A-bound complex IV may be more flexible than in the NDUFA4-bound complex. However, as the functional consequences of this difference remain unclear, we chose to moderate our interpretation. Accordingly, we revised the text and removed the concluding sentence from the Results section.

Line 360-363:

The density for Glu62 of COX2 in the HIGD2A-bound CIV suggests uncertainty regarding side chain conformation, consistent with high flexibility (Supplementary Figure 4F, H). Interestingly, this region appears slightly more ordered in the presence of NDUFA4 (Supplementary Figure 5G; I), indicating a potential stabilizing effect.

3) The functional relevance of HIGD2A as a placeholder for NDUFA4 is unclear.

The primary previously studied function of HIGD2A is its involvement in COX3 module biogenesis⁹⁻¹¹.

In the Discussion, we speculate that HIGD2A may act as a placeholder to maintain the complex in a partially assembled state until the appropriate NDUFA4 isoform is incorporated. This hypothesis is supported by the structural and biochemical evidence suggesting that HIGD2A and NDUFA4 are inserted consecutively.

Interestingly, both HIGD2A and the NDUFA4L2 isoform are known to be regulated by oxygen levels. This raises the possibility that HIGD2A plays a regulatory role in adapting complex IV assembly to oxygen availability.

We acknowledge that this proposed new function warrants further investigation, particularly under defined oxygen conditions and with isoform-specific analyses.

We modified the text to highlight that the functional relevance needs to be further investigated.

Line 505-508:

Further investigation will be required to explore the mechanistic link between either NDUFA4 or HIGD2A incorporation, K-pathway stabilization, and efficient oxygen reduction in both isolated CIV and the respirasome.

More details on my main concerns, as well as some minor comments, are included in the text below.

Detailed comments

In line 36, “SC assembly” should be replaced by “the canonical respirasome assembly”, as the findings of the paper only pertain to this supercomplex.

We changed it. Thank you for pointing it out.

Line 36-38:

These structures, along with biochemical data, reveal that the canonical respirasome assembly concludes with the final maturation of CIV in association with mature CI and CIII₂.

Lines 40 to 43: unclear. Which binding factors are the authors referring to here? None seems to be mentioned in the manuscript.

It refers in this case to the different isoforms of NDUFA4, as we mention in the Discussion. For clarity, we have now replaced the sentence with:

Line 41-45:

By revealing the temporal dynamics of complex IV assembly, our data suggest that placeholders like HIGD2A prevent premature recruitment of the final components of the complexes, NDUFA4 or its isoforms, and act as molecular clocks to ensure the correct progression of pre-SC particles into fully functional respirasomes.

Line 54: the reducing equivalents that fuel the MRC are contained in NADH and succinate, not NADH and FADH2. FADH2 is a cofactor of CII, analogous to FMN in CI, while NADH and succinate are the reduced species that donate electrons to CI and CII, respectively.

We changed the text to clarify.

Line 53-56:

The mitochondrial respiratory chain (MRC) consists of four multimeric enzymatic complexes (CI to CIV) and two mobile electron carriers, ubiquinone (Q) and cytochrome c (cyt c), which act in concert to drive the transfer of electrons from substrates such as NADH and succinate, to molecular oxygen.

Lines 128-129: where is COX14 in the structures? Since it is used for affinity purification, it would be expected to find it in the structures of the purified material. It appears that COX14 is an assembly factor involved in the biogenesis of the COX1 module, so it is not clear how it would be bound to/allow purification of late assembly intermediates and fully assembled supercomplexes. The authors should clarify this aspect. It is possible that the observed structures derive from the extremely abundant respirasomes present in the mitochondrial membranes, but this would mean the structures come from “contaminants” of the purification and this should be stated clearly. The current wording is misleading, as it suggests that tagging COX14 was necessary to purify the solved structures. If instead this was indeed the case, the authors should explain how COX14 would allow the purification of fully assembled and late assembly respirasomes.

As explained in our response to the previous comment (1c), we have revised the text to avoid confusion. We would like to emphasize that reciprocal IP analyses using HIGD2A and NDUFA4 as baits (Figure 1) detected COX14, albeit only in small amounts, suggesting that the protein is still present during the late stages of respirasome assembly.

Lines 139-140: where are the results of 3D variability analysis of CIV shown?

We now added a Supplementary Video 1 to show the movement of Complex IV relative to Complex I and III. Briefly, to prove the conformational movement of Complex IV, we fixed the Complex I position and used the mask covering Complex III and Complex IV for 3D variability analysis (cryoSPARC). The 3D variability display job was used to create a volume series of 20 intermediate reconstructions that express the conformational variability of Complex III and Complex IV relative to Complex I. While the relative position of Complex III is almost stable, Complex IV adopts a variable conformation. This observation indicated the conformational flexibility of Complex IV.

Lines 248-251: based on the provided map (HIGD2A_bound_complexIV_map.ccp4), it does not appear that most side chains from residue 31 to 106 show unambiguous density. As commented below for figure S1, it appears that the reported resolutions don't match the quality of the maps and the peripheral region of CIV, where HIGD2A is located, is strongly affected

by the issue. While there is density for some bulky residues (such as Phe39, Tyr62, Arg80) and the overall topology is in good agreement with the alphafold prediction provided by UniProt (<https://www.uniprot.org/uniprotkb/Q9BW72/entry#structure>), which point towards the density being indeed HIGD2A, “with sufficient resolution to unambiguously resolve most amino acid side chains from Asn31 to Pro106 at the distal C-terminal domain” is an overstatement.

As mentioned in the previous comment, we agree that the local resolutions at peripheral regions including NDUFA4 and HIGD2A are lower than overall resolution indicated in the text and Supplementary Figure. We now reported the maps showing local resolutions in Supplementary Figure 2 and changed the text accordingly. We also removed word “unambiguous”. (Lines 260-262)

Lines, 302-304: as above, the resolution of the map region corresponding to NDUFA4 appears to be worse than 3.2 Å. Although the agreement with the previously reported structure (PDB 5Z62) and the good side chain density for some bulky residues (such as Phe19 or Tyr 32) support the density assignment, which is hereby not disputed, the wording in the text is misleading. I suggest removing this sentence.

Thank you for raising this point. We rephrased this sentence as below:

Line 325-328:

Although the local resolution of the map covering the NDUFA4 region is lower than 3.2 Å, ranging from 4 to 6 Å (Supplementary Figure 2, class 4 of complex IV), clear side chain densities are visible for several bulky residues, including Phe19, Tyr32 and Trp45.

Lines 258-270: it would be good to add to Fig S5 a comparison of the CIV structures from yeast CIII2CIV/CIII2CIV2 featuring Rcf and the human CIV featuring HIGD2A from this study, to make it easier to follow the paragraph. Also, in line 263 the authors refer to Rcf1, while in the rest of the paragraph they refer to Rcf2, so Rcf1 appears to be a mistake.

We thank the reviewer for this helpful suggestion. The mention of Rcf1 in previous line 263 was correct. While the nomenclature of HIG domain-containing proteins can be confusing, our intent was not to imply strict homology between yeast and human proteins. Rather, Rcf1 and HIGD2A appear to fulfill functionally similar roles in the assembly of Complex IV, particularly in stabilizing early intermediates involving the COX3 module, as shown in recent studies^{9,10}. By contrast, yeast Rcf2 functions later in respiratory chain biogenesis, associating with mature CIV and CIII₂-CIV supercomplexes, where it regulates CIV activity rather than promoting its assembly. This regulatory role of Rcf2 therefore seems more analogous to that of human HIGD1A. Nevertheless, sequence analyses indicate that both Rcf1 and Rcf2 share similarity with HIGD2A, especially through the conserved QRRQ motif (Supplementary Figure 6B). Structural comparisons between Rcf2-bound yeast SC (PDB: 6T0B) and CIV (PDB: 8C8Q) (Supplementary Figure 6C) reveal similar binding positions. However, these similarities do not necessarily imply overlapping functional roles for these proteins in mitochondrial respiratory chain organization.

We have modified the text to clarify this:

Line 276-284:

This interaction with mature CIV closely resembles that of HIGD2A, even though Rcf2 is described as modulating CIV function rather than its assembly¹², and it is considered the

functional homolog of human HIGD1A, a type-1 HIG1 family member (Supplementary Figure 6A-B). Although HIGD2A's functional homolog is Rcf1, it has been suggested that Rcf1 binds CIV in a similar manner than Rcf2¹³, which may explain the parallels between HIGD2A binding to CIV described here and the previously characterized interaction of Rcf2 with CIV, despite their distinct functions. Nonetheless, some differences are observed between HIGD2A and Rcf2 binding to CIV...

Lines 332-334: based on the provided maps, the whole TMH2 of COX2 does not show very high-resolution density, therefore making side chain-based conclusions (shown in Fig 3C and S4 F-G) appears to be an overstatement. What appears to be data-supported is that NDUFA4 stabilizes TMH2 of COX2 compared to the HIGD2A-featuring CIV, but if the rationale provided in lines 327-328 is followed and higher flexibility is necessary for proton uptake, it is then unclear why an assembly intermediate (CIV featuring HIGD2A) would be more "proton uptake ready" than the fully assembled complex (CIV featuring NDUFA4). Related to this, the discussion (lines 459-463) should be edited accordingly.

As mentioned above, we have updated Supplementary Figure 5 to include panels S5H and S5I, which present the B-factors of the COX2 protein in both HIGD2A- and NDUFA4-bound complex IV. We agree with the reviewer that, while we wish to report these observations, it would not be appropriate to base strong claims on subtle structural differences. As this evidence is not particularly strong, we have therefore moderated the results section (as mentioned above) and our discussion.

Line 501-508

Our structures suggest the possibility that NDUFA4 binding may influence proton transfer by stabilizing the entry point of the K-pathway. In this scenario, oxygen reduction at the CIV catalytic center may proceed efficiently only in the fully mature holoenzyme, possibly linking NDUFA4 binding to the functional activation of CIV, both on its own and within the respirasome. Further investigation will be required to explore the mechanistic link between either NDUFA4 or HIGD2A incorporation, K-pathway stabilization, and efficient oxygen reduction in both isolated CIV and the respirasome.

Line 338: a comment remained in the text
Removed.

Lines 338-339: Ref 53 already showed that NDUFA4 is a subunit of CIV, so while it is true that this study confirms the fact, I suggest the previous paper be cited here.
Added.

Line 340: the authors do not provide any evidence, apart from the structure-based speculation, that NDUFA4 regulates proton transfer by stabilizing the entry point of the K-pathway. I suggest rephrasing to "NDUFA4 binding might regulate proton transfer", or providing biochemical evidence and/or references from other studies that this is indeed the case.

Lines 315-341: based on my comments on lines 332-334 and 340, I suggest the authors revise this section, alongside Figs 3 and S4, by (1) limiting the discussion to secondary structure-level conformational changes of COX2 TMH2, which are well supported by the data and (2) clarifying the implications of a NDUFA4-mediated stabilization of COX2 TMH2 for the proton transfer.

We agree with the reviewer and have moderated the results section and discussion (as mentioned above).

Figure 4D, there is no signal for HIGD2A in the WT in the respirasome area of the gel, making it difficult to conclude that overexpression of NDUFA4 reduces HIGD2A binding to CIV (lines 393-397), with relation to the respirasome assembly cited in line 404.

We thank the reviewer for indicating this point. Detecting a strong HIGD2A signal in supercomplexes (SCs) by BN-PAGE is technically difficult, as this protein is present at low levels in these structures. Considering that only ~30% of total CIV in the mitochondrial respiratory chain is incorporated into SC I+III₂+IV¹⁴ (DOI: 10.1074/jbc.M106474200), and only a fraction of that has HIGD2A bound, detecting or quantifying the effect of NDUFA4 overexpression on this specific structure by BN-PAGE is particularly challenging. We often need to overexpose the film to begin seeing respirasome-bound HIGD2A, which results in high background and overly intense signals in other bands, limiting interpretability. Nevertheless, we were able to detect the respirasome-bound HIGD2A in two of the three experiments we performed (**Figure R1**), although these images are not of enough quality to be published.

Importantly, HIGD2A stabilization in the absence of NDUFA4 has been observed both in isolated CIV and in respirasome-bound CIV. Therefore, although the displacement of HIGD2A upon NDUFA4 overexpression is only biochemically evident in isolated CIV, we hypothesize that the same mechanism may also occur within the supercomplex.

Figure R1: Distribution of HIGD2A in CIV and CIV-containing supercomplexes analyzed by BN-PAGE of isolated mitochondria from WT cells overexpressing HIGD2A (2A), NDUFA4 (A4), or carrying an empty vector (EV) in two independent experiments. Green arrows indicate HIGD2A signal in SC I+III₂+IV_n.

Also, in Figure 4F there seems to be less NDUFA4 in the respirasome of HIGD2A-KO+NDUFA4 than in HIGD2A-KO+EV, but looking at the level of NDUFA4 in monomeric CIV or the ATP5A loading control it does not seem like the difference would be ascribed to loading different amount of material: what would be the explanation for this?

We thank the reviewer for this insightful comment. We quantified the three blue native immunoblot experiments and calculated the average NDUFA4 signal relative to WT in both holo-CIV and SCs I+III₂+IV_n. Based on this analysis, we did not detect statistically significant differences between the HIGD2A+EV and HIGD2A+A4 groups (**Figure R2**). The differences initially observed in Figure 4F may have resulted from experimental variability, such as uneven

antibody or ECL incubation across the membrane. We have therefore replaced the NDUFA4 blot in Figure 4F with a more representative example.

Figure R2: NDUFA4 distribution in SC I+III₂+IV_n and CIV in *HIGD2A*-KO. Distribution of NDUFA4 in CIV and CIV-containing supercomplexes analyzed by BN-PAGE of isolated mitochondria from WT and *HIGD2A*-KO cells overexpressing *HIGD2A* (2A), NDUFA4 (A4), or carrying an empty vector (EV) in three independent experiments. The NDUFA4 signal in both SC I+III₂+IV_n and holo-CIV was normalized by ATP5A signal and plotted in the bar graph relative to WT (mean ± SD of 3 independent experiments).

Lines 417-423: there are no bar graphs in the figure, but indeed it would be good to see them.

We thank the reviewer for noticing this mistake. We have added the bar graph to Figure 4B showing *HIGD2A* signal quantification in SCs, CIV and Sub CIV in NDUFA4-KO cells relative to WT cells.

Lines 459-463: can the authors point to activity measurements proving that in absence of NDUFA4 oxygen reduction is less efficient?

We agree with the reviewer on the relevance of analyzing CIV activity in the absence of NDUFA4. Accordingly, we performed spectrophotometric measurements of CIV activity and oxygen consumption analyses in NDUFA4-KO cell lines. The results of these experiments have been included in the revised manuscript as **Figure 3D–F**. We have also added the corresponding description and interpretation of these new data in the Results and Discussion sections. Additionally, we included a new section in the Methods to detail the protocols used for these experiments.

Line 366-372

To evaluate the functional consequences of NDUFA4 incorporation into CIV, we assessed its enzymatic activity in *NDUFA4*-KO cells. Consistent with previous reports, the absence of NDUFA4 led to an approximately 50% reduction in CIV activity, as determined by both spectrophotometric assays (**Figure 3D**) and in-gel activity staining (**Figure 4H**). This reduction correlated with a comparable (~50%) decrease in endogenous respiration in *NDUFA4*-KO cells relative to WT HEK293T cells (**Figure 3E, F**).

Lines 464-466: how is the data helping understand the underlying pathogenic mechanisms of Leigh syndrome? What kind of information are the current structures adding to the state of the art? It appears that references 68 and 69 concern patients in which NDUFA4 is absent, or almost absent, so how are the presented data helping to explain the pathogenic mechanisms of Leigh syndrome?

We appreciate the reviewer's observations and have revised the text to clarify how our data contribute to the understanding of NDUFA4 disruption-associated forms of Leigh syndrome. The following paragraph has been included in the updated version of the manuscript:

Line 509-515:

NDUFA4 mutations have been identified in patients suffering from Leigh syndrome associated with CIV deficiency. Our structural and biochemical data provide a mechanistic basis whereby *NDUFA4* loss disrupts the final maturation of CIV, impairing its activity and mitochondrial respiration without necessarily abolishing respirasome assembly. By delineating the temporal sequence and regulation of CIV assembly, our findings establish a direct link between structural perturbations and the molecular pathogenesis of mitochondrial disorders.

Figure S1: The particle numbers don't add up. The consensus refinement indicates 301k particles, from which the local refinements are performed. But the total of the four CIV classes is around 334k particles, so it is not clear where the additional particles come from. Additionally, it appears that only a subset of the particles were used to refine CI (168k particles) and CIII (213k particles). In the Methods section, the authors state that "the focus mask further removed the poor alignment particles", but this does not seem very likely because in the respirasome CI and CIII usually drive the alignment, partly because the membrane arm of CI and the proximal protomer of CIII form the "rigid core" of the respirasome and also because the eight FeS clusters of CI and the six hemes of CIII provide strong densities that aid the particle alignment. It is therefore strange that CI and CIII would be poorly aligned in the respirasome, particularly CI which "loses" almost half of the particles from the consensus refinement to the focused refinement. As there is no data shown for the processing steps leading to the final maps of CI and CIII, one cannot see how the removed classes look like, or at which stages they were removed. It is therefore important that the authors show exactly how the data was processed for CI and CIII. Since so many particles are removed, is it possible that other intermediates, perhaps lacking the N-module of CI, are present in the dataset and were discarded as "poorly aligned"? Or are the discarded particles corresponding to the open state of CI? Also, what is the overlap between discarded particles for CI and CIII? Meaning, are the discarded particles coming from the same respirasome particles? If so, is it possible that those are generally damaged particles, which therefore should not be considered for CIV refinement either? When focused refinements are run starting from a consensus map, to boost the resolution, one would expect that the number of particles does not change because this is the pool that corresponds to the structure of interest, in this case the respirasome. Removing one

third (as was done for CIII) or half of the particles (as was done for CI) is unexpected and appears unjustified, therefore needs to be clarified and supported by data.

Thank you for these useful suggestions. We have now carefully reviewed data processing and updated Supplementary Figure 1 with more information. The “poorly alignment dataset” is not a correct term and has been removed. In the revised Supplementary Figure 1, we show the 4 classes of complex I and III after 3D classification by using the focus masks. For complex I, the two classes display better structural features corresponding to a higher resolution map. Two classes were then combined for CTF refinement, local refinement, followed by structural modeling. After combining two classes, we now have 236,346 particles of complex I for the final map. This map has been deposited as EMD-54784. We also update it in the data availability part of the manuscript.

Same for complex III, the unused particles and classes after 3D classification display worse structural features than the main class (213,731 particles). Therefore, we kept using this main class for the final map and structural modeling.

We have modified and added following sentences in the Method part of the manuscript to clarify:

Line 745-752:

For CI and CIII refinement, the 3D classification using the focus mask (Supplementary Figure 1) further discarded the subsets of particles displaying minor populations, resulting in 236,346 particles and 213,731 particles, respectively, for the final refinement. It should be noted that the unused particles exhibited lower resolution structural features or appeared to represent minor conformational states. These were not further analyzed in this study to focus on predominant structural states as the conformational change analysis of CI and CIII have been intensively studied previously^{40,42-44,85,86}.

Lines 676-678: Along the same line, in the methods section it is stated that the CIV focused classification was done with 6 classes, then the 6 classes should be shown in the supplementary figure, also indicating which classes were pooled based on their overlapping features.

After carefully reviewing the data processing, we corrected the number of classes to eight, and the maps have been shown in the updated Supplementary Figure 1.

Figure S1 lacks:

- 1) the map-to-model FSC curves for each modelled structure
- 2) the half-map FSC curve for the consensus map
- 3) the local resolution depictions (i.e. surface colored by local resolution), as well as
- 4) the angular distribution plots for each focused map and the consensus map.

The description of how the plots and colored maps are obtained should then also be added to the methods section. The methods section should also state whether the maps were sharpened and if so how.

Points 3 and 4 are particularly important because the provided maps don't look like they correspond to the reported resolutions, so the local resolution coloring would clarify which regions of the maps really reach the reported resolution and the angular distribution would shed light on a potential anisotropy issue, which might contribute to explaining why the maps don't look like they correspond to the reported resolution.

As the “class 2” and “class 3” complex IV maps provided don't look anisotropic, it might

also be that the weird appearance of the ccp4 maps is a sharpening artifact, which is why it would be important to declare whether the maps were sharpened.

Thank you for indicating these important points on data processing and refinement process. The FSC curve for the consensus map is included in Supplementary Figure 1. We have now made a new Supplementary Figure 2 with Gold standard Fourier shell correlation (GSFSC) curves, the angular distribution plots, the local resolution depictions for all 6 maps that we processed. We also added map-to-model FSC curves for 4 modeled structures.

Four maps, including maps for CI, CIII, class 1, and class 4 of CIV, were sharpened using phenix.auto_sharpen tool in Phenix. These sharpened maps were used for further model building. While maps for class 2 and class 3 were kept as original maps from the local refinement job.

We have added this information in the Method section.

Line 710-712:

The final maps of CI, CIII, class 1 and class 4 of CIV from local refinement jobs were subjected to autosharpen job in Phenix v1.21.1, and the output maps were used for model building.

Minor comments

Figure S4: in panels F and G, the subunit to which Y244 and K319 belong (COX1 presumably) should be indicated in the legend.

Information has been added to the figure and figure legend.

All figures displaying densities (Figs. 1, S2, S3, S4) should indicate the contour used in the legend.

All figures displaying densities now contain the contour level indicated in each panel or figure legend.

Line 604: The section is titled “immunoprecipitation”, but lines 606-628 feature the description of sample purification for cryo-EM. I suggest this section is thus separated and titled “cryo-EM sample preparation”

We agree and as suggested by the reviewer, a new section “Cryo-EM sample preparation” (Line 685) has been added and separated from the previous one.

Legend to Fig S2: state that NDP stands for NADPH, as this is not clarified.

This sentence is updated in the figure legend of Supplementary Figure 3 (Supplementary Supplementary Figure 2 in the previous version).

Electrons are transferred from NADPH (NDP) to the flavin mononucleotide (FMN) in the hydrophilic domain and then sequentially down a series of Fe-S clusters (orange-yellow spheres).

Fig 2D: the figure, or legend, lack reference to the scalebar used to calculate the electrostatic potential.

Added in line 316-318:

Electrostatic surface potentials are colored red and blue for negative and positive charges, respectively, and white color represents neutral residues.

Reviewer #2 (Remarks to the Author):

The assembly of the respiratory chain and supercomplex formation plays important roles in human health and disease. This paper describes a cryo-EM study of human respirasomes, purified using an engineered tag on a complex IV subunit. The analysis revealed I2III2IV1 supercomplexes with high-resolution structures of fully assembled CI and CIII, while CIV density is weak. Local refinement and classification of the flexibly attached CIV yielded distinct classes, containing either the assembly factor HIGD2A and/or the subunit NDUFA4, which is the last subunit to be added to the complex.

In this interesting paper the authors make a convincing case that the removal of HIGD2A is necessary for insertion of NDUFA4, as their positions partially overlap. However, the authors do not mention what, if any, role HIGD2A plays in respirasome formation. Also, it is not clear if acting as a “placeholder” for NDUFA4 is the only function of HIGD2A or if it plays a role in the attachment of the COX3 module, as previously suggested. The paper would be strengthened if these issues were addressed.

We thank the reviewer for their positive and constructive comments.

Our study builds on previous biochemical work that clearly established HIGD2A’s role in COX3 insertion. This function was carefully analyzed in earlier studies⁹⁻¹¹, and our current structural findings show close interaction between HIGD2A and COX3, which supports and is consistent with this proposed role.

We acknowledge that structural analysis of a HIGD2A KO would further confirm its function in COX3 insertion. However, designing such a study poses significant technical challenges. Loss of HIGD2A would likely affect the early stages of complex IV assembly, potentially resulting in unstable or incomplete assembly intermediates—even in the context of the supercomplex—making such an experiment hard to visualise structurally. While this remains an important avenue for future work, it is beyond the scope of the current study.

Instead, we focused on a novel and unexpected observation: the dynamic interplay between HIGD2A and NDUFA4. To determine the sequence of their insertion and their interdependence, we performed additional analyses using KO models, confirming that NDUFA4 functionally replaces HIGD2A during complex IV maturation.

In the revised Discussion, we have highlighted previous studies describing the relationship between HIGD2A and COX3. We also emphasize the speculation that HIGD2A may act as a placeholder until the appropriate NDUFA4 isoform is incorporated. This could be particularly relevant under varying physiological conditions, as both HIGD2A and NDUFA4 isoforms are regulated by oxygen availability.

Regarding the functional relevance of our results to respirasome formation, we are happy to report that our data support the notion that complex IV (CIV) assembly occurs within the context of the whole respirasome. However, we currently lack direct evidence on how disruption of HIGD2A or NDUFA4 affects the assembly of the other components of the respirasome. We therefore consider this an important avenue for future investigation.

Specific comments:

1. The protein was purified with a tag on CIV, so presumably yielding both respirasomes and single CIV. However, the latter is never mentioned. Are the particles visible in the micrographs? If so, why are these not analysed? If not, how can this be explained?

We thank the reviewer for this important point. During data processing, we specifically searched for particles corresponding to Complex IV alone. However, we did not obtain any 2D class averages that resembled isolated CIV. Initially, we attempted to use material from FLAG co-immunoprecipitation directly for grid preparation and collected a small dataset, but the resulting sample was highly heterogeneous, and subsequent 2D classification failed to yield useful classes. To improve sample quality, we optimized the preparation workflow. In particular, we included additional post-immunoprecipitation centrifugation steps, including a 16-hour spin at 55,000 rpm in a TLA55 rotor, which enriched for higher-molecular-weight assemblies while depleting smaller complexes. We believe that this step substantially reduced the presence of isolated CIV, thereby explaining their absence in our cryo-EM data. This choice was made to minimize heterogeneity and maximize the recovery of well-resolved respirasome particles for high-resolution structural analysis. Details of these steps are provided in the *Methods* section under "Sample preparation for cryo-EM".

2. The introduction, p. 4 states "it has been proposed that HIGD2A promotes the formation of respirasome assembly intermediates by coordinating the association of CIV assembly modules, revealing divergent assembly pathways for individual and supercomplexed CIV". Is there any evidence for this? For a distinction it would be necessary to study individual CIV as well. It is nowhere shown how CIV is oriented in the respirasome and what subunits interact with CI or CIII.

Recently, there has been increasing evidence that the formation of the respirasome happens through a cooperative assembly mechanism. Regarding the binding of CIV precursors to SCs, biochemical analyses in CIV-deficient cell lines showed that SC CIII2 is stabilized through its interaction via assembly factor HIGD2A with an atypical MT-CO3 module bound to MT-CO1^{10,15,16}.

We agree with the reviewer that it is necessary to study individual CIV and how HIGD2A affects its assembly. However, we were unable to visualize CIV-like particles, despite many efforts in particle picking and 2D classification. As mentioned in the previous comment, the sample of COX14-FLAG coimmunoprecipitation has been investigated biochemically, showing interactions with individual and complexed CIV; however, our cryoEM failed to identify CIV like particles.

We added Supplementary Video 1 to show the flexibility of CIV relative to CI and CIII complex as analyzed by 3D variability. On the other hand, the orientation of CIV relative to CI and CIII had been well studied in the previous reports of the mammalian respirasomes¹⁷⁻²⁰.

3. Further to that, the model in Fig. 5 does not show what induces CIV attachment to the respirasome.

We thank the reviewer for this insightful question. Two models of respirasome assembly have been proposed: (1) the plasticity model, in which individual complexes assemble first and then associate into supercomplexes; and (2) the cooperative model, where supercomplexes form through the association of incomplete intermediates^{16,21}. The coexistence of free MRC

complexes and respirasome is demonstrated by both reported high-resolution structures and blue native-PAGE (BN-PAGE). Recently, there has been a growing body of evidence suggesting binding of CI, CIII₂ and CIV precursors before the maturation of the individual complexes, which corresponds to the second model^{15,22,23}. Capturing such intermediate states at high resolution remains technically challenging due to their low abundance and transient nature, although biochemical approaches such as BN-PAGE provide indirect support. Regarding the binding of CIV precursors to SCs, recent data, particularly from CIV-deficient cell lines, suggest that an atypical COX3 module, stabilized by HIGD2A, can associate with CI–CIII₂, forming a potential precursor (“CI–CIII₂ plus”) prior to full CIV maturation^{10,15}. While our assembly model in Fig. 5 is inspired by this hypothesis, we refrain from specifying the precise sequence of COX module incorporation, as our data do not directly capture intermediate states. To avoid overinterpretation, we grouped the three COX modules together and clearly stated in the legend that their assembly order remains unresolved.

4. p. 12: The story about the flexibility at the K-pathway entrance is confusing. The region is said to typically have high B factors, indicating flexibility, nevertheless distinct conformations are assigned to these uncertain densities. Particularly suspicious is Glu62: a negatively charged side chain typically has no cryo-EM density (unless the charge is compensated by a nearby positive charge like a salt bridge or ion ligand) and the side chain can't be modeled correctly even in a rigid structure. The absence of the glutamate sidechain is obvious in Suppl. Fig. 4F and G (although just showing density for one residue does not give an indication of the level of flexibility in this region). Thus, the suggestion that the conformation of Glu62 in a bovine CIV structure in GDN represents an intermediate state between the two human structures is a very weak argument (Fig. 3C). If the authors feel that there is a real conformational change in this region, better figures will be needed showing the actual densities, not just models. Otherwise, this section needs to be toned down. Further to suppl. fig. 4F,G: please indicate which subunits the highlighted residues belong to. Glu62 is misspelled as Glu61 in the legend.

We thank the reviewer for raising this point. Both reviewers expressed concern about this section, and we agree that our original description may have been an overinterpretation. Of note, we have updated Supplementary Figure 5 (previously Figure S4) to include panels S5H and S5I, which present the B-factors of the COX2 protein in both HIGD2A- and NDUFA4-bound complex IV. The entrance of the K-pathway, particularly Glu62, displayed a higher B factor in the HIGD2A-bound state compared to the NDUFA4-bound state. However, as this evidence is not sufficiently strong and we don't provide functional relevance, we have moderated our discussion and removed the sentence from the results: 'Additionally, it suggests that NDUFA4 binding may regulate proton transfer through the K-pathway.'

Line 501-508

Our structures suggest that NDUFA4 binding may influence proton transfer by stabilizing the entry point of the K-pathway. In this scenario, oxygen reduction at the CIV catalytic center would proceed efficiently only in the fully mature holoenzyme, linking NDUFA4 binding to full functional activation of CIV, both as an individual complex and within the respirasome. Further investigation will be required to define the mechanistic link between NDUFA4 or HIGD2A incorporation, K-pathway stabilization, and efficient oxygen reduction in isolated CIV and in the respirasome.

5. Figure 1B: please indicate how CIV is oriented relative to the respirasome shown in 1A.

The orientation of complex IV shown in Figure 1B is the same as the left one in Figure 1A. We now updated the figure legend.

6. Cryo-EM methods (p.23):

a. What is the protein concentration?

We added this information to ‘Sample preparation for cryoEM’ part in the Methods.

Line 708-709

Grids were prepared using protein at a concentration of 1 mg/ml.

b. What microscope was used?

We have now added this information to ‘Cryo-EM data acquisition and data processing’ in the Methods.

Line 718-721

All cryo-EM data collection (Supplementary Table 1) was performed with EPU (Thermo Fisher Scientific) operated on Titan Krios G3i transmission electron microscope (Thermo Fisher Scientific) at 300 kV in the Karolinska Institutet’s 3D-EM facility.

c. Please add “total electron dose” of 35 e-/Å².

Thank you! Added.

d. How many particles were picked?

We added this information to ‘Cryo-EM data acquisition and data processing’ part in the Methods.

Line 731:

A total of 2,508,182 particles were initially picked from two data sets.

e. “bad alignment particles” and “poor alignment particles”: please rephrase.

We have updated this information in ‘Cryo-EM data acquisition and data processing’ part in the Methods.

Line 738-742:

After homogeneous refinement of these particles, 3D classification without mask followed by iterative heterogeneous refinements was performed to get rid of classes displaying broken densities, resulting in 336,847 particles yielding 3.1 Å resolution after homogeneous refinement (Supplementary Figure 1).

Line 744-752:

Masked refinements of the respective SC intermediates resulted in map resolutions of 2.9 Å for CI, 2.8 Å for CIII and 3.1 Å for CIV. For CI and CIII refinement, the 3D classification using the focus mask (Supplementary Figure 1) further removed the subsets of particles displaying minor populations, resulting in 236,346 particles and 213,731 particles,

respectively, for the final refinement. It should be noted that the unused particles exhibited lower resolution structural features or appeared to represent minor conformational states. These were not further analyzed in this study to focus on predominant structural states as the conformational change analysis of CI and CIII have been intensively studied previously^{40,42-44,81,82}.

f. What is a “local refinement volume”?

Answer:

We are sorry for causing confusion. The focus masks are described in ‘Cryo-EM data acquisition and data processing’ part in the Methods and Supplementary Figure 1.

Line 745-748:

For CI and CIII refinement, the 3D classification using the focus mask (Supplementary figure 1) further removed the subsets of particles displaying minor populations, resulting in 236,346 particles and 213,731 particles, respectively, for the final refinement.

g. Global or local CTF refinement?

We performed both global and local CTF refinement. We have now updated the Method section with this information.

7. Model building (p. 24):

a. Densities for lipids were manually fitted using references: please explain.

We carefully inspected the lipid molecules in the previously reported models of complex I, III and IV [PDB: 8OM1, 5XTD for CI; 5XTE, 8UGD for CIII; 5Z62, 8UGL for CIV] and used them as references to fit the lipids into our maps. We now add reference for each PDB structure.

b. “The ligands, cofactors and modifications were manually inspected as suggested in Uniprot and experimentally determined structures reported previously”. Unclear.

During refinement of each protein or subunit as components of CI, CIII and CIV, we carefully inspected each protein one by one by using the sequence and binding cofactors, which have been listed in the database in Uniprot. On the other hand, several previously reported models of complex I, III and IV have been inspected as reference models.

We now rephased this sentence for clarification:

Line 780-783:

The ligands, cofactors and modifications of proteins or subunits in individual complexes were manually inspected according to Uniprot database and experimentally determined structures reported previously (PDB: 8OM1 for CI; 8UGD for CIII; and 5Z62 for CIV).

c. HIGD2A was manually built with an AlphaFold structure as reference. Please remove “manually”.

Removed.

8. Suppl. fig. 1: The workflow needs more details/images: exemplary micrographs, 2D class averages, number of particles picked initially, initial models.

Answer:

We now updated Supplementary Figure 1 and added new Supplementary Figure 2, which together contain more details about data processing and refinement.

References:

- 1 Kampjut, D. & Sazanov, L. A. The coupling mechanism of mammalian respiratory complex I. *Science* **370**, eabc4209, doi:doi:10.1126/science.abc4209 (2020).
- 2 Chung, I. *et al.* Cryo-EM structures define ubiquinone-10 binding to mitochondrial complex I and conformational transitions accompanying Q-site occupancy. *Nat. Commun.* **13**, 2758, doi:10.1038/s41467-022-30506-1 (2022).
- 3 Gu, J., Liu, T., Guo, R., Zhang, L. & Yang, M. The coupling mechanism of mammalian mitochondrial complex I. *Nat. Struct. Mol. Biol.* **29**, 172-182, doi:10.1038/s41594-022-00722-w (2022).
- 4 Chung, I., Grba, D. N., Wright, J. J. & Hirst, J. Making the leap from structure to mechanism: are the open states of mammalian complex I identified by cryoEM resting states or catalytic intermediates? *Curr. Opin. Struct. Biol.* **77**, 102447, doi:<https://doi.org/10.1016/j.sbi.2022.102447> (2022).
- 5 Xia, D. *et al.* Structural analysis of cytochrome bc1 complexes: Implications to the mechanism of function. *Biochim. Biophys. Acta* **1827**, 1278-1294, doi:<https://doi.org/10.1016/j.bbabi.2012.11.008> (2013).
- 6 Sarewicz, M. *et al.* Catalytic Reactions and Energy Conservation in the Cytochrome bc1 and b6f Complexes of Energy-Transducing Membranes. *Chem. Rev.* **121**, 2020-2108, doi:10.1021/acs.chemrev.0c00712 (2021).
- 7 Richter-Dennerlein, R. *et al.* Mitochondrial Protein Synthesis Adapts to Influx of Nuclear-Encoded Protein. *Cell* **167**, 471-483.e410, doi:10.1016/j.cell.2016.09.003 (2016).
- 8 Weraarpachai, W. *et al.* Mutations in C12orf62, a Factor that Couples COX I Synthesis with Cytochrome c Oxidase Assembly, Cause Fatal Neonatal Lactic Acidosis. *The American Journal of Human Genetics* **90**, 142-151, doi:10.1016/j.ajhg.2011.11.027 (2012).
- 9 Hock, D. H. *et al.* HIGD2A is Required for Assembly of the COX3 Module of Human Mitochondrial Complex IV. *Mol. Cell. Proteomics* **19**, 1145-1160, doi:10.1074/mcp.RA120.002076 (2020).
- 10 Timón-Gómez, A., Garlich, J., Stuart, R. A., Ugalde, C. & Barrientos, A. Distinct Roles of Mitochondrial HIGD1A and HIGD2A in Respiratory Complex and Supercomplex Biogenesis. *Cell Reports* **31**, doi:10.1016/j.celrep.2020.107607 (2020).
- 11 Timón-Gómez, A., Bartley-Dier, E. L., Fontanesi, F. & Barrientos, A. HIGD-Driven Regulation of Cytochrome c Oxidase Biogenesis and Function. *Cells* **9**, 2620 (2020).
- 12 Römpler, K. *et al.* Overlapping Role of Respiratory Supercomplex Factor Rcf2 and Its N-terminal Homolog Rcf3 in *Saccharomyces cerevisiae*. *J. Biol. Chem.* **291**, 23769-23778, doi:10.1074/jbc.M116.734665 (2016).
- 13 Moe, A., Ädelroth, P., Brzezinski, P. & Näsvik Öjemyr, L. Cryo-EM structure and function of *S. pombe* complex IV with bound respiratory supercomplex factor. *Communications Chemistry* **6**, 32, doi:10.1038/s42004-023-00827-3 (2023).
- 14 Schägger, H. & Pfeiffer, K. The Ratio of Oxidative Phosphorylation Complexes I–V in Bovine Heart Mitochondria and the Composition of Respiratory Chain

- Supercomplexes*. *J. Biol. Chem.* **276**, 37861-37867, doi:<https://doi.org/10.1074/jbc.M106474200> (2001).
- 15 Lobo-Jarne, T. *et al.* Multiple pathways coordinate assembly of human mitochondrial complex IV and stabilization of respiratory supercomplexes. *EMBO J.* **39**, e103912, doi:<https://doi.org/10.15252/emj.2019103912> (2020).
- 16 Fernández-Vizarra, E. & Ugalde, C. Cooperative assembly of the mitochondrial respiratory chain. *Trends in Biochemical Sciences* **47**, 999-1008, doi:<https://doi.org/10.1016/j.tibs.2022.07.005> (2022).
- 17 Gu, J. *et al.* The architecture of the mammalian respirasome. *Nature* **537**, 639-643, doi:10.1038/nature19359 (2016).
- 18 Letts, J. A., Fiedorczuk, K. & Sazanov, L. A. The architecture of respiratory supercomplexes. *Nature* **537**, 644-648, doi:10.1038/nature19774 (2016).
- 19 Wu, M., Gu, J., Guo, R., Huang, Y. & Yang, M. Structure of Mammalian Respiratory Supercomplex I_{III}II_{IV}1. *Cell* **167**, 1598-1609.e1510, doi:10.1016/j.cell.2016.11.012 (2016).
- 20 Guo, R., Zong, S., Wu, M., Gu, J. & Yang, M. Architecture of Human Mitochondrial Respiratory Megacomplex I_{II}III_{II}IV₂. *Cell* **170**, 1247-1257.e1212, doi:10.1016/j.cell.2017.07.050 (2017).
- 21 Vercellino, I. & Sazanov, L. A. The assembly, regulation and function of the mitochondrial respiratory chain. *Nature Reviews Molecular Cell Biology* **23**, 141-161, doi:10.1038/s41580-021-00415-0 (2022).
- 22 Protasoni, M. *et al.* Respiratory supercomplexes act as a platform for complex III-mediated maturation of human mitochondrial complexes I and IV. *EMBO J.* **39**, e102817, doi:<https://doi.org/10.15252/emj.2019102817> (2020).
- 23 Fang, H. *et al.* A membrane arm of mitochondrial complex I sufficient to promote respirasome formation. *Cell Reports* **35**, doi:10.1016/j.celrep.2021.108963 (2021).

Response to reviewers' comments

Reviewer #1 (Remarks to the Author):

I thank the authors for thoroughly considering and addressing my comments. I am satisfied with the revised version of the manuscript and recommend publication without further revision.

We thank the reviewer for their evaluation and are pleased that our revisions have addressed their comments.

Reviewer #2 (Remarks to the Author):

The authors have properly addressed most of my comments. One issue remains: the interpretation of differences in the K pathway between complex IV maps in the presence of NDUFA4 or HIGD2A. Line 360: "The density for Glu62 of COX2 in the HIGD2A-bound CIV suggests uncertainty regarding side chain conformation, consistent with high flexibility (Supplementary Figure 5F). Interestingly, this region appears slightly more ordered in the presence of NDUFA4 (Supplementary Figure 5G), indicating a potential stabilizing effect.". In the revised manuscript the authors have added local resolution plots, which clearly show that the resolution in this region is poor in both maps. But in fact, looking at the maps provided by the authors, both CIV maps also suffer from anisotropic resolution, which is not visible from a static picture. Due to all these factors, even the backbones can't be reliably interpreted in either map, let alone sidechains. And, as stated in my original review, negatively charged sidechains typically have no cryo-EM density. So the "uncertainty regarding side chain conformation" is not even due to the low resolution. Further, the authors state that "this region appears slightly more ordered in the presence of NDUFA4 (Supplementary Figure 5G), indicating a potential stabilizing effect". Sfig. 5 F and G show density for Glu62 only, in 5G at a very low contour level where there is a blob of density assigned to the sidechain. The figure is missing all the context of the maps, where there is no indication of "better ordering" in one of them.

Figure 3C shows the "conformational change" of Glu62, including a previously published structure (8D4T from EMD-27196). I checked that map and although it has better resolution than the ones from the present study, there is no density for the (charged) Glu62 sidechain and its conformation in the structure is based on guesswork.

In short, the authors do not provide any evidence for a conformational change or lower flexibility due to NDUFA4 binding. The region is simply flexible in all structures.

It would be best if the whole story (line 340-364) is eliminated, including Figure 3C and Sfig. 5F and 5G.

We agree with the reviewer that this was the most speculative part of the manuscript, and we have removed it from the final version.

Figure 3C, Supplementary Figures 5F and 5G, and the corresponding text (lines 340–364) have been deleted. We also removed Supplementary Figures 5H and 5I, which presented subtle differences in B-factor values for the COX2 protein in the HIGD2A-bound and NDUFA4-bound complex IV.

Additionally, we revised the corresponding section of the Discussion (previously lines 501–508) as follows:

Previous text:

The incorporation of NDUFA4 is synchronized with the release of HIGD2A from CIV, completing the assembly of both isolated and supercomplexed CIV. Our structures suggest the possibility that NDUFA4 binding may influence proton transfer by stabilizing the entry point of the K-pathway. In this scenario, oxygen reduction at the CIV catalytic center would proceed efficiently only in the fully mature holoenzyme, linking NDUFA4 binding to full functional activation of CIV, both as an individual complex and within the respirasome. Further investigation will be required to define the mechanistic link between NDUFA4 or HIGD2A incorporation, K-pathway stabilization, and efficient oxygen reduction in isolated CIV and the respirasome.

Revised text:

The incorporation of NDUFA4 is synchronized with the release of HIGD2A from CIV, completing the assembly of both isolated and supercomplexed CIV. It is tempting to speculate that NDUFA4 binding may influence proton transfer. In this scenario, oxygen reduction at the CIV catalytic center would proceed efficiently only in the fully mature holoenzyme, linking NDUFA4 binding to full functional activation of CIV, both as an individual complex and within the respirasome. Further investigation will be required to define the mechanistic link between NDUFA4 or HIGD2A incorporation and efficient oxygen reduction in isolated CIV and the respirasome.